# Beyond Benchmarks: Understanding Mixture-of-Experts Models through Internal Mechanisms

## Abstract

Mixture-of-Experts (MoE) architectures have emerged as a promising direction, offering efficiency and scalability by activating only a subset of parameters during inference. However, current research remains largely performance-centric, with limited understanding of its internal mechanisms, thereby constraining broader progress. In this work, we use an internal metric to investigate the mechanisms of MoE architecture by explicitly incorporating routing mechanisms and analyzing expert-level behaviors. Through systematic analyses of a wide range of publicly available MoE models, we uncover several findings: (1) neuron utilization decreases as models evolve, reflecting stronger generalization; (2) training exhibits a dynamic trajectory, where benchmark performance alone provides limited signal while MUI reveals deeper insights; (3) task completion emerges from collaborative contributions of multiple experts, with shared experts driving concentration; and (4) activation patterns at the neuron level provide a fine-grained proxy for data diversity. Together, these results demonstrate the potential of MUI as a complementary indicator to benchmark performance, offering new insights into the capacity, dynamics, and specialization of MoE models.

## 1 Introduction

With the rapid advancement of Large Language Models (LLMs), an increasing number of Mixture-of-Experts (MoE) architectures have been proposed, such as DeepSeek (Liu et al., 2024), GPT-OSS (OpenAI, 2025), and Qwen3 (Yang et al., 2025). Unlike dense models that rely on fixed forward parameters, MoE models employ dynamic routing, selectively activating different subsets of parameters known as experts. This design offers two key advantages: 1) both training and inference are more efficient, as only a small fraction of parameters are activated for each input; and 2) performance can be improved, one common explanation is that, under the same computational budget, MoE models can be scaled to much larger parameter sizes. However, are these advantages the only reason behind MoE's success? Our current understanding of MoE architectures remains limited, and this lack of interpretability poses challenges for their further development.

From these perspectives, it is essential to investigate the mechanisms of MoE architectures. Current research primarily focuses on benchmark performance for understanding, but benchmark performance alone is insufficient. As LLMs increasingly saturate widely used benchmarks, the performance differences across models become marginal, while potential benchmark leakage (Zhou et al., 2023; Ying et al., 2024a) further undermines the reliability of these results. In this work, we use an internal metric to investigate the mechanisms of MoE architectures, extending the Model Utilization Index (MUI) originally proposed on dense models Cao et al. (2025), which measures the proportion of neurons required for task completion. However, unlike dense models, MoE requires explicitly accounting for routing mechanisms when evaluating the degree to which a model utilizes its internal capacity. Moreover, it is equally important to enable fine-grained investigations at the expert level in order to better understand the functional roles and contributions of individual experts. Through systematic analyses of a wide range of publicly available MoE models, and by tracing how internal mechanisms evolve as model capabilities change, we not only demonstrate the applicability of our adapted indicators but also conduct in-depth analyses at the expert-level. Based on these analyses, we uncover several findings and provide the following key insights:

1. Reduced neuron utilization with model evolution: within the same family, we observe that as models evolve, their performance improves while requiring fewer neurons to accomplish the same tasks. We believe this is a reflection of stronger generalization. Notably, GPT-OSS models exhibit strikingly low MUI, which may explain why GPT-5 achieve superior performance in real-world applications — strong generalization (Section 3.2).

2. Dynamic MUI trajectory during training: by tracking how MUI evolves throughout the training process, we provide insights beyond performance metrics, showing how MUI can serve as an indicator for monitoring training dynamics and guiding model development (Section 3.3).

3. Collaborative expert contributions: task completion often emerges from the joint collaboration of multiple experts. Stronger models exhibit a higher proportion of expert cooperation, with GPT-OSS showing the highest. Interestingly, the presence of shared experts further drives expert concentration, potentially diminishing the diversity advantages of distributed experts (Section 3.4).

4. Data measurement: activation patterns at both the neuron and expert levels reflect data diversity, while neuron-level offering a more efficient way (Section 3.5).

## 2 MoE Model Utilization Index

To address limitations in performance-only evaluation, we propose designing internal indicators that monitor MoE models from the perspective of underlying mechanisms. Building on recent advances in interpretability, which have shown how model components, such as neurons and layers, interact to produce overall behavior (Pan et al., 2024; Cao et al., 2025), we extend this line of inquiry to MoE architectures. Specially, we focus on identifying the proportion of key neurons required for task completion, and study how this proportion evolves during capability shifts and across training iterations. These dynamics form the basis of a meaningful measure for model monitoring, which we term the MoE-MUI (MUI for simple). To this end, we first introduce the neuron importance calculation method primarily used in our study in the following section. It is important to note that our proposed metric is not tied to any specific interpretation method. To ensure robustness, we further consider several alternative formulations, which are detailed in our ablation studies (Section 4).

### 2.1 Preliminary

Neuron-level interpretable methods (Dai et al., 2022a; Geva et al., 2021) connect individual neurons in the feed-forward network (FFN) sub-layer of LLMs to specific semantic meanings. These neurons can be treated as mediator variables (Meng et al., 2022) for certain model behaviors. In MoE models, each expert $\mathbf{E}_i$ corresponds to one FFN. Specifically, omitting the layer normalization for brevity, the MoE layer $l$ can be defined as a function of input hidden state $\mathbf{x}^l$:

$$\text{MoE}^l(\mathbf{x}^l) = \sum_{i \in \mathcal{R}(\mathbf{x}^l)} \mathbf{G}_i^l(\mathbf{x}^l)\mathbf{E}_i^l(\mathbf{x}^l) + \sum_{s \in \mathcal{S}} \mathbf{G}_s^l(\mathbf{x}^l)\mathbf{E}_s^l(\mathbf{x}^l),$$

$$\mathbf{E}_i^l(\mathbf{x}^l) = \left(\mathbf{x}^l \mathbf{W}_{\text{u},i}^l \odot (\mathbf{x}^l \mathbf{W}_{\text{g},i}^l)\right)\mathbf{W}_{\text{d},i}^l,$$

(1)

where $\mathcal{R}(\mathbf{x}^l)$ is the routed (top-$k$) expert set, $\mathcal{S}$ is the shared expert set, $\mathbf{G}_i^l(\mathbf{x}^l)$ are routing weights (for shared experts, set $\mathbf{G}_s^l(\mathbf{x}^l) \equiv 1$ if they are always active), and $\mathbf{W}_{\text{u},i}^l, \mathbf{W}_{\text{g},i}^l, \mathbf{W}_{\text{d},i}^l$ are the projections in SwiGLU. Following (nostalgebraist, 2020), for $j$-th neuron in expert $i$ at layer $l$ contributing to prediction of token $\hat{y}$ when given input sequence $x$, we define the token-level neuron contribution:

$$f_{\text{neuron}}(i, j, l, \hat{y} \mid x) = \left(\mathbf{G}_i^l(\mathbf{x}^l) \cdot (\mathbf{x}^l \mathbf{W}_{\text{g},i}^l) \cdot \mathbf{W}_{\text{d},i}^l\right)[j] \cdot \mathbf{W}_{\text{head}}[:, \hat{y}],$$

(2)

where $\mathbf{W}_{\text{head}}$ is the unembedding matrix mapping hidden states to vocabulary logits. For a given threshold $\eta$, the key activated neurons for task sample $s = (x, y)$ is defined as:

$$N_{\text{activated}}(s) = \left\{(i, j, l) \mid \exists \hat{y}_t, f_{\text{neuron}}(i, j, l, \hat{y}_t \mid x \oplus \hat{y}_{<t}) > \eta\right\},$$

(3)

where $\hat{y}_t$ denotes the $t$-th token in $y$, $\hat{y}_{<t}$ denotes the partial output sequence before the $t$-th token, and $\oplus$ represents concatenation with the input sequence.

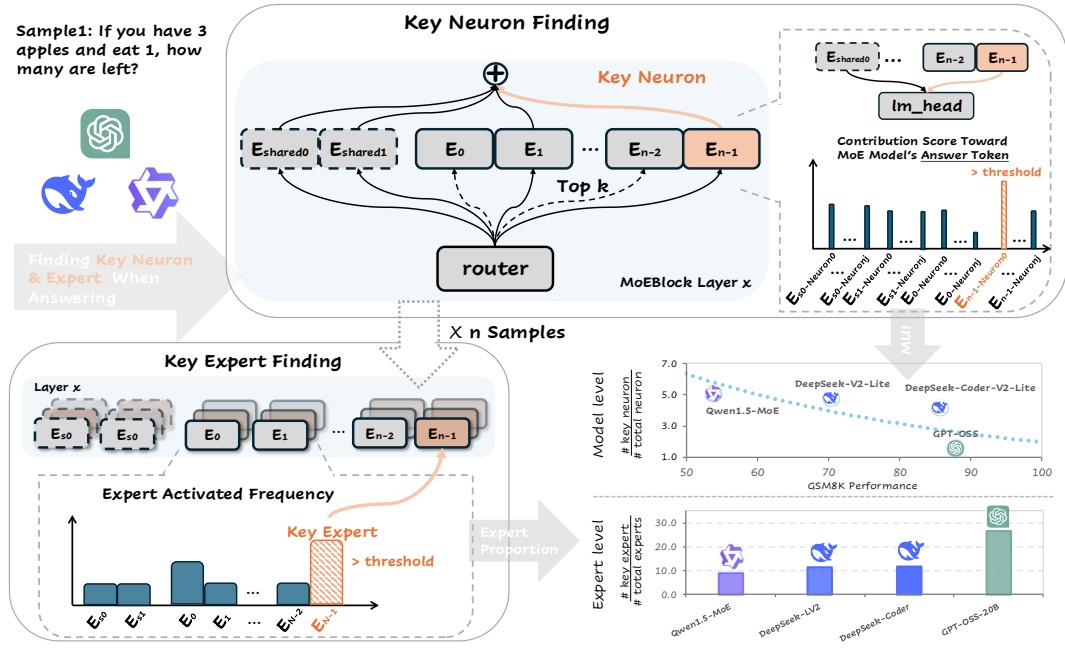

Figure 1: Illustration of getting key neurons and key experts for given task samples ($s_1$ to $s_n$).

## 2.2 MODEL UTILIZATION INDEX

Given a task set $\mathcal{T} = \{s_1, s_2, \ldots, s_k\}$, we first identify the set of key activated neurons for each sample $s_i$ using Equation 2. By accumulating across all samples, we obtain the union of neurons required to complete the task. The neuron-level MUI for MoE is then defined as the proportion of activated neurons relative to the total available neurons in the model:

$$\textbf{MUI}(\mathcal{T}) = \frac{|\bigcup N_{\text{activated}}(s_i)|}{N \times L \times (|E_s| + |E_r|)}, \tag{4}$$

where $N$ is the number of neurons per expert, $L$ is the number of MoE layers, $|E_s|$ is the number of shared experts, and $|E_r|$ is the number of routed experts per layer. Correspondingly, if we focus only on the expert information contained in the activated neuron set we can identify the experts set $E_{\text{activated}}(s) = \left\{ (i, l) \;\middle|\; \exists j, (i, j, l) \in N_{\text{activated}}(s) \right\}$ that are responsible for sample $s$. Given a frequency threshold $\eta_{\text{expert}}$, we could find the set of key experts for a task set $\mathcal{T}$ as those experts that consistently contribute across samples:

$$E_{\text{key}}(\mathcal{T}) = \left\{ (i, l) \;\middle|\; \frac{\left| \left\{ s \in \mathcal{T} \,\middle|\, (i, l) \in N_{\text{activated}}(s) \right\} \right|}{|\mathcal{T}|} \geq \eta_{\text{expert}} \right\}. \tag{5}$$

Meanwhile, by aggregating the sets of task-responsible experts, we can derive both the overall proportion of key experts within the model as well as the MUI for each individual expert. Formally, the proportion of key experts for a given task $\mathcal{T}$ is defined as:

$$\text{KeyExpertProportion}(\mathcal{T}) = \frac{|\bigcup E_{\text{key}}(\mathcal{T})|}{L \times (|E_s| + |E_r|)} \tag{6}$$

In addition, for a specific expert $(i', l')$, its MUI with respect to task $\mathcal{T}$ is computed as:

$$\textbf{MUI}_{(i', l')}(\mathcal{T}) = \frac{|\bigcup \{ j \,|\, (i', j, l') \in N_{\text{activated}}(s) \}|}{N} \tag{7}$$

Figure 1 provides an illustration of our methodology. Starting from a given sample (*e.g.*, sample 1), we identify the key neurons that contribute to the model's response during inference. By aggregating results from multiple samples, we can identify the corresponding task-level experts (shown in red).

## 3 EXPERIMENTS

In this section, we present our empirical study on a broad set of open-source MoE models. Specifically, we conduct interpretability-based analyses on 13 publicly available models ranging from 20B to 200B parameters, as well as 10 intermediate checkpoints from the OLMoE series. By monitoring how the MUI changes alongside changes in model capabilities, we aim to reveal the potential of MUI as an internal indicator of model capacity. Furthermore, we demonstrate how MUI enables fine-grained expert-level analysis, offering insights into the internal dynamics of MoE architectures.

### 3.1 SETUP

**Dataset Selection.** To ensure reliable conclusions, we adopt a diverse set of widely used benchmarks. Following (Cao et al., 2025; Ying et al., 2024b), our evaluation covers three categories: 1) GSM8K (Cobbe et al., 2021), MATH (Hendrycks et al., 2021), and ARC-Challenge (Clark et al., 2018) for math reasoning, 2) HumaEval (Chen et al., 2021) and MBPP (Austin et al., 2021) for coding, 3) BIG-bench Hard (BBH) (bench authors, 2023) and MMLU (Hendrycks et al., 2020) to cover general tasks. Statistical result for the selected benchmarks is shown in Table 2.

**Model Selection.** To maximize the applicability of MUI and ensure fairness in evaluation, we select four widely used series of open-source LLMs: 1) GPT Series: GPT-OSS-20B and GPT-OSS-120B (OpenAI, 2025). 2) Qwen Series: Qwen1.5-MoE (Team, 2024), Qwen3-30B, Qwen3-Coder-30B, Qwen3-235B-Thinking (Yang et al., 2025), and Qwen3-Next. 3) DeepSeek Series: DeepSeek-MoE (Dai et al., 2024), DeepSeek-V2-Lite (abbreviated as DeepSeek-LV2), DeepSeek-Coder-V2-Lite, DeepSeek-V2, and DeepSeek-Coder-V2 (et al., 2024). 4) OLMoE Series: several checkpoints from OLMoE-7B (et al., 2025); detailed checkpoint information is provided in Appendix A.2.

**Implementation.** Details of the response generation parameters for each model are provided in the Appendix. For the threshold $\eta$ in Equation 3, we set it to the top 1‰ of total neurons, applied at the layer level (additional implementation details are reported in the Appendix A.3). Furthermore, we discuss the neuron selection strategy and justify the choice of threshold in our Section 4.

### 3.2 REDUCED NEURON UTILIZATION WITH MODEL EVOLUTION

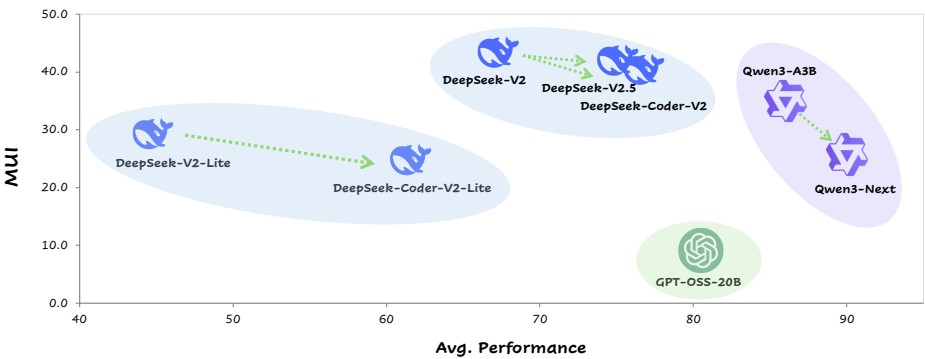

Figure 2: Overall weighted-average performance (%) and MUI (%) across selected benchmarks.

By comparing earlier and later versions within the same model families, we examine how MUI reflects the impact of model iteration or evolution. Specifically, within the DeepSeek family, we compare DeepSeek-V2 and DeepSeek-V2-Lite with their enhanced counterparts, DeepSeek-Coder-V2/DeepSeek-V2.5 and DeepSeek-Coder-V2-Lite. Although labeled as "Coder" versions, model reports (Zhu et al., 2024) and benchmark results indicate that these are a comprehensive evolution of the V2 series (MUI changes under specific capabilities improvement will be discussed in Section 3.3). Similarly, in the Qwen family, we compare Qwen3-30B-A3B with its iterative successor, Qwen3-Next. The performance and corresponding MUI values (Equation 4) for these models are shown in Figure 2. Analyzing all neurons jointly, we observe that later-released models consistently achieve stronger performance on the same datasets while exhibiting lower MUI. If we assume that these newer models indeed possess higher true capability and stronger generalization (i.e., the ability

to handle a broader range of tasks beyond the specific evaluation sets), then MUI may serve as an indicator of intrinsic capacity and generalization rather than benchmark-specific performance. This interpretation is supported by two pieces of evidence. First, prior work on dense models Cao et al. (2025) reached a similar conclusion, showing that lower MUI correlates with stronger generalization (Cao et al., 2025). Second, Team (2023) found that with increased training data, parameters become more specialized even in a single ReLU output model, while generalization simultaneously improves. Notably, GPT-OSS exhibits strikingly low MUI, which may explain why the GPT series provides consistently strong user experience in real-world — superior generalization capability.

> **MUI as an Indicator**
>
> Combining performance with MUI offers an indicator of a model's underlying generalization capability, mitigating the risks of misleading evaluations caused by leakage.

### 3.3 DYNAMIC MUI TRAJECTORY DURING TRAINING

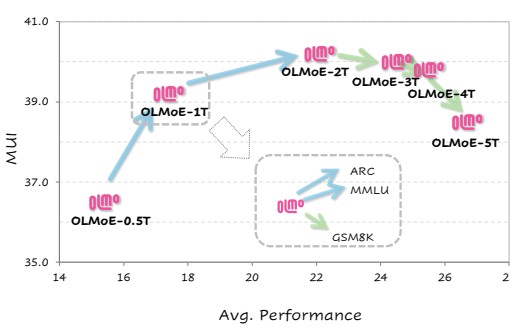
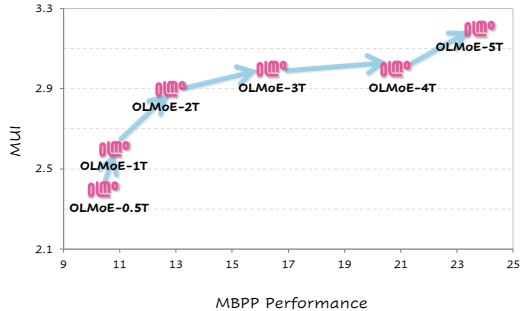

Figure 3: MUI (%) and Performance (%) change across OLMoE checkpoints trained with 0.5T, 1T, 2T, 3T, 4T and 5T tokens on the select 7 benchmarks.

Figure 4: MUI (%) and Performance (%) change across OLMoE checkpoints trained with 0.5T, 1T, 2T, 3T, 4T and 5T tokens on the benchmark MBPP.

Previous results indicate that later-stage models achieve lower MUI alongside improved performance. However, an important question remains: does MUI decrease monotonically throughout training, or do different phases exhibit distinct trajectories? To address this, we monitor MUI for the fully open-source OLMoE models across the entire training process, with the goal of deriving insights that can inform training strategies and model development. Figure 3 plots the overall performance across seven selected tasks alongside the corresponding MUI values for each checkpoint, with more detailed statistics reported in Table 5. The results reveal a two-phase trajectory in training. At earlier stages, performance improvements are accompanied by an increase in MUI, which we refer to as the "Accumulating" phase. In this phase, the model appears to recruit a larger set of neurons for memorization and learning (Team, 2023). This trend also emerges when models undergo capability-specific improvements. For example, compared to DeepSeek-V2, DeepSeek-Coder-V2 places greater emphasis on coding ability. As a result (Table 4), its MUI increases on coding tasks such as MBPP (from 4.9 to 6.3) and HumanEval (from 2.7 to 3.3).

At later stages, however, further performance gains occur together with a decrease in MUI, which we call the "Evolving" phase. This suggests that with continued exposure to more data, the model transitions toward more efficient utilization. As indicated by our earlier analysis in Section 3.2, such efficiency is closely associated with improved generalization. Importantly, this dynamic learning trajectory is not uniform across all capabilities but results from a mixture of ability-specific trends. For instance, as shown in Figure 3, after training on 1T tokens, OLMoE-1T exhibits an Evolving trend on GSM8K, whereas ARC and MMLU continue to follow the Accumulating trajectory. These heterogeneous ability-specific patterns collectively determine whether the model's overall trajectory appears Accumulating or Evolving at the aggregate level. Thus, we summarize our takeaway as:

> **Takeaway: MUI Moniting MoE Training**
>
> Monitoring performance alone is insufficient; MUI provides a complementary perspective for performance for detecting divergent trajectories and adjusting training accordingly.
> For example, as shown in Figure 4, in coding tasks such as MBPP, OLMoE consistently remains in the Accumulating phase without entering the Evolving phase. This suggests that additional coding data, or a higher proportion of coding tasks during earlier training stages, may be required to help the model further improve its generalization ability.

## 3.4 COLLABORATIVE EXPERT CONTRIBUTIONS

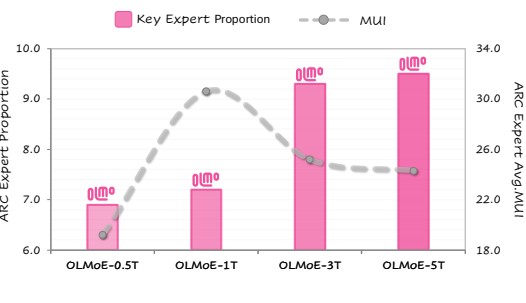

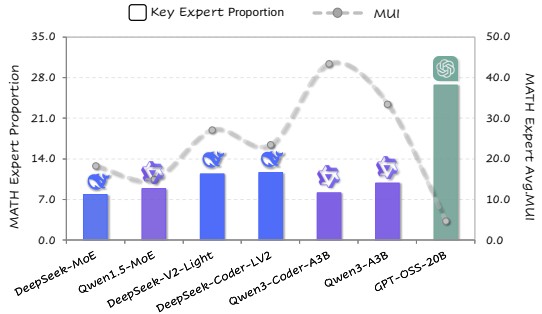

Figure 5: Proportion of key experts for the ARC task and the corresponding MUI within those key experts across the OLMoE series.

Figure 6: Proportion of key experts for the Task task and the corresponding MUI within those key experts across the selected MoE models.

After establishing the potential of neuron-level activation as an indicator of model capacity, we now extend our analysis to the expert level. Specifically, following Equation 5, we examine how experts contribute to completing the task. For a given task, the distribution of activated experts can be viewed as a probability distribution over the expert set. To quantify expert contributions, we adopt a frequency threshold of $\eta_{\text{expert}} = 0.6$ to identify key experts that only with consistently involved. While detailed activation distributions for each benchmark are presented in Figures 12 through 16. Considering that different architectures employ varying numbers of experts, we focus here on models within the same architecture family for comparison. As shown in Figure 5, 1). OLMoE exhibits an increasing proportion of key experts (Equation 6) as training progresses (solid bar), with more consistent results reported in Table 6 through Table 9. With GPT-OSS consistently demonstrates the highest proportion of key experts among the models studied. 2). At the same time, the MUI within these key experts (Equation 7) shows a trajectory that first rises and then falls. That is, during early training, the model recruits a large number of neurons (reflected by increasing MUI). As training progresses, specialization emerges, potentially consolidated within experts, leading to a compression phase where MUI decreases. At the same time, having a broader set of experts in "Collaboration", results in stronger overall performance and improved generalization. This suggests that for **MoE models, activating a larger number of experts while requiring fewer neurons within each expert is often associated with stronger true capability and better generalization.** This observation is consistent with the pattern shown in Figure 6, where GPT-OSS exhibits a markedly different trajectory from other models — aligning well with our hypothesis above.

After analyzing the overall trend of expert utilization, we further analyze the distribution of key experts, particularly in light of architectural differences between shared and routed experts. As shown in Figure 7, we present the results on the MMLU dataset, with additional results provided in Appendix B.3. For MoE architectures that include shared experts, the findings reveal that the top-10 most frequently activated experts are exclusively shared experts. Moreover, the utilization rate of these shared experts is extremely high; for instance, in Qwen3-Next, each shared expert is activated in more than 90% of the cases. By contrast, in GPT-OSS, a routed-only MoE, the activation rate is extremely low. Considering the training dynamics of the two types of experts — shared experts

being persistently active versus routed experts only being activated when selected by the router — this implicates how experts emerge differently across MoE architectures:

> **Expert "Collaboration" in MoE**
>
> Though exhibiting an increasing trend of expert "collaborative" during training. In models having shared experts, their persistent activation leads to concentrated responsibility within the shared pool, whereas in routed-only architectures, the influence of load-balancing losses drives a more dispersed "many-hands" collaboration among a broader set of experts.

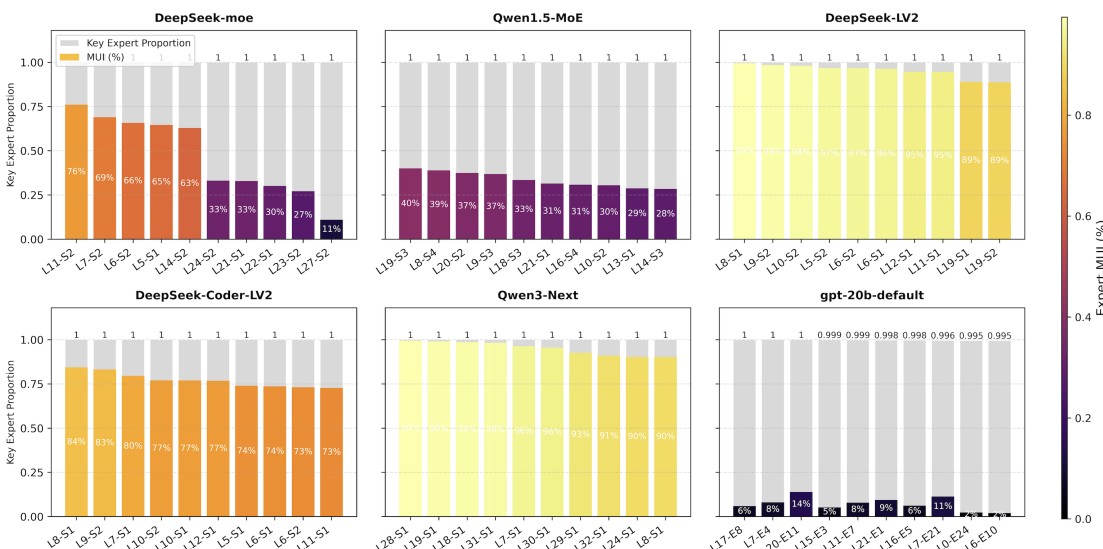

Figure 7: Top-10 experts (ranked by activation frequency in Equation 5) for the selected MoE models with shared-expert structures (the exception GPT-OSS-20B model is included for comparison) on MMLU. The corresponding MUI for each expert is reported. Shared experts are denoted as $S_i$.

Given that the proportion of shared experts is relatively small, **task-responsible experts tend to be disproportionately concentrated within the shared pool.** To test this hypothesis, we examine whether shared experts are enriched in the intersection of per-task key experts. Specifically, we evaluate three benchmarks from different domains and report in Table 1. Fisher's exact tests reveal highly significant enrichment, with odds ratios ranging over $54.6$ and two-sided $p$-values from $10^{-29}$ to $10^{-57}$. This indicates that shared experts are not only frequently activated within individual tasks but also disproportionately dominate the set of experts consistently activated across multiple tasks. In other words, shared experts act as common capacity hubs that concentrate responsibility across tasks, confirming our hypothesis that training tends to centralize task responsibility within this subset. Notably, Qwen3-Next shows the lowest proportion of key experts, which can be attributed to its architecture containing the smallest shared-expert ratio (1 out of 513 experts). This concentration has a dual implication. On the one hand, shared experts provide "capacity hubs" that can enable efficient cross-task knowledge transfer. On the other hand, such reliance risks over-centralization, potentially limiting expert specialization and reducing the effective diversity of the expert pool.

| Task | DeepSeek-MoE | Qwen1.5-MoE | DeepSeek-V2-Lite | DeepSeek-Coder-LV2 | Qwen3-Next |
|---|---|---|---|---|---|
| GSM8K | 8.3 / 35.6 | 10.2 / 57.1 | 11.9 / 25.4 | 14.1 / 21.5 | 2.8 / 7.0 |
| GSM8K + MBPP | 4.8 / 61.4 | 6.6 / 82.3 | 5.9 / 51.5 | 7.9 / 38.5 | 0.6 / 31.3 |
| GSM8K + MBPP + MMLU | 3.5 / 84.4 | 6.0 / 90.3 | 3.3 / 91.2 | 5.8 / 52.0 | 0.4 / 49.0 |

Table 1: Key experts proportion(%) / proportion of shared experts among key experts(%).

In summary, although the feed-forward networks in MoE architectures are referred to as "Experts," it is difficult in practice to interpret them as independent task-specific units, whether in routed-only designs or in ones that also include shared experts. In routed-only architectures, the presence

of load-balancing losses prevents learning from consistently concentrating on specific experts, and continued training instead yields experts with sparse and diffuse specialization. As a result, it is difficult to establish a stable union of task-responsible experts that could facilitate further analysis. In architectures with shared experts, task responsibility tends to converge largely within the shared pool, leading to substantial overlap in the experts activated across different tasks. In some sense, this behavior resembles that of dense models, but it also risks undermining the diversity benefits that multiple experts are expected to provide.

### 3.5 DATA MEASUREMENT

Since the activation of different neurons or experts corresponds to engaging distinct regions of the model's internal capacity, these activation patterns can be leveraged as an internal proxy for measuring the diversity of input data. To illustrate this, we conduct experiments by randomly sampling data from the three selected domains. In addition, considering the potential influence of reasoning length, we divide each domain into two subsets: the top 50% longest samples (denoted as long) and the remaining shorter samples (denoted as short). The results in Figure 8 and Figure 25 indicate: 1) both the neuron-level MUI and the proportion of activated experts ($\eta_{expert} = 0$) are positively correlated with data diversity, and this correlation remains robust regardless of input length.

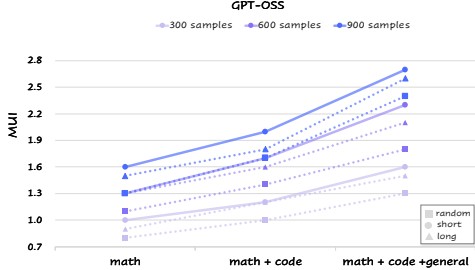

Figure 8: MUI across different diversity.

This confirms the validity of our case-level, rather than token-level, measurement strategy, as MUI is not artificially inflated by longer reasoning chains; 2) compared to neurons, expert-level activation yields much higher ratios (typically above 90%) due to the larger parameter scale. As a result, the expert-level activation rate saturates and is less suitable for measuring diversity across datasets. In contrast, neuron-level MUI offers finer granularity and efficiency: 600 samples spanning from three domains have comparable MUI to 900 samples from a single domain.

## 4 ABLATION STUDY

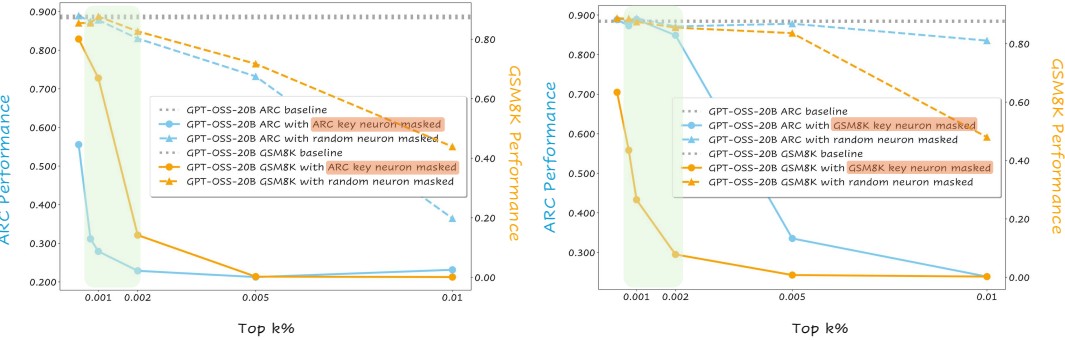

Figure 9: Performance (ACC%) of the Llama-3-8B-Instruction model on the ARC and GSM8K datasets, with key neurons masked specifically for ARC and GSM8K. Key neurons are identified using Equation 3 and a layer-level top-k threshold function (detailed in Appendix A.3). The threshold value used for our MUI analysis —1‰, is visually indicated by a green box .

The previous experiments have demonstrated both the effectiveness and the insightfulness of our proposed methodology. However, one potential concern is the validity of the identified neurons, particularly regarding the choice of thresholds in Equation 3. To address this, we employ neuron masking, a widely used intervention technique in mechanistic interpretability, to verify whether the selected neurons bear a causal relationship to the model's output. Here, we mask the neurons identified on ARC and GSM8K under different threshold values (to eliminate the confounding effect of

varying model shape, we adopt percentage-based thresholds rather than absolute counts). Ideally, the more task-relevant key neurons are masked, the more pronounced the resulting performance degradation should be. The result for GPT-OSS in Figure 9 (other shown in Figure 26) confirms our selection: The results, shown for GPT-OSS in Figure 9 (with additional results in Figure 26), confirm this expectation. First, masking neurons identified by either ARC or GSM8K produce a steady decline in performance as more neurons are removed, with a much sharper drop than under random masking. This effect is especially pronounced when $k$ lies in the range of 0.1% — 0.2% of neurons. Second, masking neurons derived from ARC using $k$ in the same interval significantly reduces performance on both ARC and GSM8K. In contrast, masking neurons derived from GSM8K has little impact on ARC performance. This arises because ARC spans a broader set of abilities beyond mathematical reasoning, whereas GSM8K primarily focuses on arithmetic reasoning. These findings further demonstrate the validity of our neuron selection approach, and our chosen threshold evidently falls within this interval, enabling us to identify task-critical neurons in a principled manner. In addition to the 0.1% threshold reported in the main paper, we also experiment with alternative thresholds, which have consistent results (Appendix C). For completeness, we further evaluate alternative importance scoring methods, and due to space constraints, we report in Appendix C.

## 5 RELATED WORK

Recent years have seen a resurgence of MoE (Cai et al., 2025; Dai et al., 2024; Jiang et al., 2024), whose core idea is to activate only a few experts per token. To stabilize training and improve specialization, DeepSeekMoE introduces always-on shared experts (Dai et al., 2024), a design later integrated into DeepSeek-V2 (et al., 2024) and DeepSeek-V3 (Liu et al., 2024) and adopted in subsequent systems such as Qwen3-Next (Yang et al., 2025).Meanwhile, a line of work analyzes and improves expert specialization and load balancing. StableMoE proposes a two-stage training strategy with router distillation to reduce routing volatility and stabilize convergence (Dai et al., 2022b). Expert-Choice Routing instead lets experts select tokens rather than tokens selecting experts, which leads to better load balancing (Zhou et al., 2022). To address the issue of experts collapsing into similar behaviors, OLMoE introduces orthogonalization and diversity-promoting regularization (et al., 2025). On the balancing side, new approaches remove the dependence on auxiliary losses (Wang et al., 2024). Beyond optimization, some works (Xue et al., 2024) conduct token-level analyses of expert. However, there is still a lack of comprehensive and in-depth analysis of MoE models.

## 6 DISCUSSION

This work represents an attempt to employ interpretability methods as tools for MoE model understanding and evaluation. Through extensive experiments, we demonstrate the promise and the practical utility of the MUI. Nonetheless, interpretability remains an evolving field, and several limitations of our study should be acknowledged. 1) while we observe clear internal transitions toward the Evolving phase across model iterations, establishing a direct and quantitative link between MUI & performance and generalization remains challenging. 2) our implementation is grounded in neuron-based interpretability techniques. Though we tested alternative formulations, the broader interpretability community continues to debate best practices and methodologies. As interpretability tools evolve, MUI itself should be revisited and refined.

## 7 CONCLUSION

In this work, we move beyond benchmark-centric evaluations and provide a deep analysis of MoE models through the lens of internal utilization patterns. Our systematic investigations further highlight that stronger models not only achieve higher performance but also have reduced neuron utilization, and more collaborative expert behaviors. These findings indicate MUI with performance serves as an indicator of both training progress and generalization strength, providing a new diagnostic tool for model development. Expert-level analyses demonstrate that MoE functionality emerges from collective expert interactions rather than isolated contributions. Further analysis shared experts showing how their dominance centralizes task responsibility. We hope these results will encourage future work to build on internal utilization analyses as a complementary perspective for understanding, improving, and controlling MoE architectures.

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

## A   EXPERIMENT DETAILS

### A.1   DATASET STATISTICAL RESULT

Following (Cao et al., 2025; Ying et al., 2024b), we focus on three representative abilities: Mathematical and Reasoning, Coding, and General Capability. For each ability, we select a set of publicly available datasets to evaluate. To balance computational cost and coverage, we sample from large-scale benchmarks such as BBH (bench authors, 2023) and MMLU (Hendrycks et al., 2020), while ensuring evaluation quality by following the sampling protocol in (Wang et al., 2025). A detailed summary of the statistical characteristics of the selected datasets is provided in Table 2.

| Model | GSM8K (Math & Reasoning) | MATH (Math & Reasoning) | ARC$_c$ (Math & Reasoning) | HumanEval (Code) | MBPP (Code) | BBH (General) | MMLU (General) | Totally |
|---|---|---|---|---|---|---|---|---|
| # Testing Samples | 1,319 | 5,000 | 1,172 | 164 | 500 | 2,000 | 4,000 | 14,155 |

Table 2: The statistical detail of the selected benchmarks.

### A.2   OLMoE SERIES MODEL SELECTION

For OLMoE (et al., 2025) series model, we include eight checkpoints detailed in Table 3.

| Custom Checkpoint Name | Original Checkpoint Name | Training Steps | Training Tokens |
|---|---|---|---|
| OLMoE-0.5T | step120000-tokens503B | 120,000 | 503B |
| OLMoE-1T | step245000-tokens1027B | 245,000 | 1,027B |
| OLMoE-1.5T | step365000-tokens1530B | 365,000 | 1,530B |
| OLMoE-2T | step490000-tokens2055B | 490,000 | 2,055B |
| OLMoE-2.5T | step610000-tokens2558B | 610,000 | 2,558B |
| OLMoE-3T | step735000-tokens3082B | 735,000 | 3,082B |
| OLMoE-3.5T | step855000-tokens3586B | 855,000 | 3,586B |
| OLMoE-4T | step975000-tokens4089B | 975,000 | 4,089B |
| OLMoE-4.5T | step1100000-tokens4613B | 1,100,000 | 4,613B |
| OLMoE-5T | step1200000-tokens5033B | 1,200,000 | 5,033B |

Table 3: Summary of the checkpoints of OLMoE used in the study. "Custom Checkpoint Name" represents simplified names defined in this paper for clarity.

### A.3   MECHANISTIC INTERPRETABILITY TECHNIQUES

For the neuron contribution score, we primarily adopt one of the most commonly used methods, as described in Section 2, for time and computational cost considerations. Nevertheless, our approach is not confined to a single interpretation technique. Our objective is to explore MUI under multiple perspectives to ensure comprehensive conclusions. As detailed in our ablation study (Section 4), we also experiment with alternative neuron analysis methods to further validate our findings.

When applying the method defined in Section 2, the response $y_t$ in Equation 3 is generated under benchmark-specific conditions. For BBH, we use the original 3-shot setting. For all other benchmarks, instruction-tuned models are evaluated in a zero-shot setting, while OLMoE models, being base models, are evaluated with a one-shot prompt. Detailed configurations of model generation, along with the few-shot examples, are provided in Appendix A.5.

The threshold $\eta$ in Equation 3 is set to the top 1‰ of neurons per layer (corresponding to the top 1‰ of $N$), thereby selecting the most salient neurons at each layer level. This threshold function is detailed as follows:

$$N_{\text{activated}}(s) = \left\{ (i,j,l) \ \middle| \ \exists \ \hat{y}_t, f_{\text{neuron}}\left(i,j,l,\hat{y}_t \mid x \oplus \hat{y}_{<t}\right) \geq V_l^{top1‰} \right\}, \tag{8}$$

where : $V_l = [f_{\text{neuron}}\left(i,j,l,\hat{y}_t \mid x \oplus \hat{y}_{<t}\right) \mid i \in \{1,2,\ldots,n\}, j \in \{1,2,\ldots,|E|\}, i \in \{1,2,\ldots,N\}]$, with $N$ representing the number of neurons in the tested model, $n$ representing the length for response $y$, and $|E|$ representing the number total experts.

### A.4 Implementation for Alternative Neuron Importance Definitions

In addition to the contribution-based method described in the main text, we also explored several alternative definitions of neuron importance. These implementations include threshold-based functions as well as other heuristic approaches for quantifying neuron contribution. Here we are also considering directly using the activation score (Marked as activate):

$$f_{\text{neuron}_{activate}}(i, j, l, \hat{y} \mid x) = \Big( \mathbf{G}_i^l(\mathbf{x}^l) \cdot (\mathbf{W}_{\text{u},i}^l \odot (\mathbf{x}^l \mathbf{W}_{\text{g},i}^l)) \Big)[j], \qquad (9)$$

where $\mathbf{G}_i^l(\mathbf{x}^l)$ are routing weights (for shared experts, set $\mathbf{G}_s^l(\mathbf{x}^l) \equiv 1$ if they are always active), and $\mathbf{W}_{\text{u},i}^l, \mathbf{W}_{\text{g},i}^l, \mathbf{W}_{\text{d},i}^l$ are the projections in SwiGLU. Moreover, in architectures that only employ a single up-projection and down-projection without a gating mechanism, the output directly maps to the vocabulary space (nostalgebraist, 2020). We adapt this formulation to the MoE setting:

$$f_{\text{neuron}_{glu}}(i, j, l, \hat{y} \mid x) = \Big( \mathbf{G}_i^l(\mathbf{x}^l) \cdot (\mathbf{W}_{\text{u},i}^l \odot (\mathbf{x}^l \mathbf{W}_{\text{g},i}^l)) \cdot \mathbf{W}_{\text{d},i}^l \Big)[j] \cdot \mathbf{W}_{\text{head}}[:, \hat{y}], \qquad (10)$$

### A.5 Model Parameter Setting

**Response Generation.** All models are evaluated with a fixed decoding temperature of 0.0. For non-reasoning models, the maximum output length is set to 1,024 tokens, while for reasoning-oriented models we follow their default maximum lengths: 16,384 tokens for Qwen3-235B and 131,072 tokens for the GPT series. For computational efficiency, we use the default reasoning effort setting, which we found to perform comparably, or in some cases, better than the "high" setting.

**Benchmark Conditions.** Generation settings vary depending on the benchmark. For BBH, we adopt the standard 3-shot prompts provided in the benchmark. For all other benchmarks, responses are generated in a zero-shot manner for instruction-tuned models, while OLMoE series models are evaluated under a human-crafted one-shot setting, following Cao et al. (2025).

All experiments are conducted on 32 NVIDIA H2000 GPUs, totaling approximately 1,536 GPU hours for the interpretation experiment.

## B More Experiment Results

### B.1 Model performance and MUI

| Model | GSM8K (Math & Reasoning) | MATH (Math & Reasoning) | ARC$_c$ (Math & Reasoning) | HumanEval (Code) | MBPP (Code) | BBH (General) | MMLU (General) |
|---|---|---|---|---|---|---|---|
| DeepSeek-MoE-A2.8 | 59.6 / 1.6 | 13.2 / 3.7 | 52.4 / 6.9 | 46.9 / 1.3 | 47.3 / 2.0 | 42.3 / 3.6 | 44.8 / 14.7 |
| Qwen1.5-MoE-A2.7B | 53.8 / 5.1 | 17.4 / 7.1 | 70.0 / 10.3 | 46.3 / 1.4 | 42.7 / 2.2 | 35.5 / 5.8 | 54.9 / 23.1 |
| DeepSeek-LV2-A2.4B | 70.4 / 4.8 | 23.1 / 7.7 | 69.2 / 10.5 | 50.0 / 1.7 | 48.3 / 2.9 | 49.4 / 5.5 | 53.4 / 23.1 |
| DeepSeek-Coder-LV2-A2.4B | 85.7 / 4.2 | 56.4 / 8.3 | 69.5 / 8.5 | 72.6 / 1.8 | 64.9 / 3.4 | 63.8 / 6.0 | 55.9 / 18.5 |
| Qwen3-Coder-A3B | 86.4 / 7.5 | 81.2 / 10.4 | 90.7 / 10.8 | 92.7 / 3.3 | 72.9 / 5.7 | 87.5 / 11.0 | 77.5 / 23.7 |
| Qwen3-A3B | 90.0 / 7.2 | 90.7 / 11.5 | 93.3 / 12.0 | 92.7 / 3.4 | 74.9 / 5.7 | 90.5 / 10.5 | 81.6 / 26.1 |
| Qwen3-Next | 93.6 / 5.6 | 92.0 / 9.1 | 92.5 / 10.3 | 94.5 / 2.0 | 80.8 / 3.6 | 93.3 / 7.8 | 84.7 / 25.7 |
| GPT-OSS-A3.6B | 87.9 / 1.6 | 74.2 / 2.5 | 88.3 / 3.2 | 84.7 / 0.8 | 70.5 / 1.3 | 80.0 / 2.4 | 80.1 / 6.8 |
| DeepSeek-V2-A21B | 91.2 / 6.3 | 43.5 / 13.2 | 90.8 / 15.9 | 76.8 / 2.7 | 64.3 / 5.3 | 80.7 / 8.7 | 75.4 / 35.4 |
| DeepSeek-Coder-V2-A21B | 95.0 / 5.9 | 67.2 / 13.7 | 91.1 / 14.4 | 82.9 / 3.3 | 70.0 / 6.3 | 84.5 / 9.6 | 75.5 / 29.8 |
| DeepSeek-V2.5-A21B | 91.4 / 6.6 | 64.5 / 12.2 | 88.4 / 15.4 | 84.8 / 2.6 | 67.1 / 4.9 | 85.6 / 8.9 | 75.2 / 33.8 |
| Qwen3-A22B | 91.4 / 6.3 | 89.2 / 9.0 | 89.3 / 11.3 | 87.8 / 2.6 | 82.2 / 4.6 | 79.8 / 7.9 | 83.1 / 24.2 |
| GPT-OSS-A5.1B | 85.7 / 4.4 | 75.9 / 6.6 | 88.9 / 8.0 | 81.1 / 1.7 | 70.1 / 2.8 | 78.0 / 6.0 | 84.5 / 17.2 |

Table 4: Performance (accuracy %) and MUI (%), as determined by neuron analysis (Equation 4) with threshold top $k = 0.1\%$

| Model | GSM8K (Math & Reasoning) | MATH (Math & Reasoning) | ARC$_c$ (Math & Reasoning) | HumanEval (Code) | MBPP (Code) | BBH (General) | MMLU (General) |
|---|---|---|---|---|---|---|---|
| OLMoE-0.5T | 2.8 / 7.4 | 3.3 / 11.9 | 25.7 / 10.8 | 8.5 / 1.6 | 10.4 / 2.4 | 25.7 / 8.1 | 27.2 / 25.0 |
| OLMoE-1T | 3.6 / 10.5 | 3.2 / 12.9 | 33.6 / 15.3 | 6.1 / 1.6 | 10.8 / 2.6 | 25.3 / 8.3 | 32.2 / 26.3 |
| OLMoE-1.5T | 3.8 / 10.4 | 3.8 / 10.9 | 42.2 / 17.3 | 8.0 / 1.7 | 10.6 / 2.6 | 25.6 / 8.7 | 41.3 / 28.0 |
| OLMoE-2T | 4.4 / 7.9 | 4.1 / 12.5 | 47.7 / 15.9 | 8.0 / 1.8 | 12.8 / 2.9 | 27.7 / 9.2 | 42.1 / 27.9 |
| OLMoE-2.5T | 4.7 / 9.5 | 4.0 / 11.9 | 50.8 / 15.9 | 11.6 / 1.9 | 17.4 / 3.1 | 26.2 / 8.3 | 44.0 / 26.4 |
| OLMoE-3T | 6.4 / 10.1 | 4.2 / 10.6 | 53.8 / 15.9 | 8.0 / 1.8 | 16.4 / 3.0 | 29.5 / 8.6 | 46.3 / 28.1 |
| OLMoE-3.5T | 6.4 / 10.0 | 4.5 / 13.1 | 54.5 / 15.6 | 9.1 / 1.8 | 16.0 / 2.9 | 20.0 / 9.0 | 46.7 / 27.4 |
| OLMoE-4T | 5.0 / 9.2 | 4.6 / 11.8 | 57.6 / 16.0 | 11.0 / 1.9 | 20.8 / 3.0 | 30.7 / 8.6 | 47.5 / 28.2 |
| OLMoE-4.5T | 4.4 / 6.5 | 4.7 / 10.5 | 56.7 / 15.3 | 11.6 / 1.8 | 18.0 / 3.0 | 31.8 / 8.5 | 48.5 / 28.0 |
| OLMoE-5T | 6.0 / 6.2 | 4.9 / 10.7 | 60.5 / 15.2 | 14.6 / 2.0 | 23.8 / 3.2 | 30.2 / 8.5 | 50.1 / 28.0 |

Table 5: Performance (accuracy %) and MUI (%), as determined by neuron analysis (Equation 2) with top $k = 0.1\%$

## B.2 EXPERT LEVEL ANALYZE RESULT

| Model | GSM8K (Math & Reasoning) | MATH (Math & Reasoning) | ARC$_c$ (Math & Reasoning) | HumanEval (Code) | MBPP (Code) | BBH (General) | MMLU (General) |
|---|---|---|---|---|---|---|---|
| DeepSeek-MoE-A2.8 | 59.6 / 8.2 | 13.2 / 7.8 | 52.4 / 9.1 | 46.9 / 9.9 | 47.3 / 9.3 | 42.3 / 5.7 | 44.8 / 7.6 |
| Qwen1.5-MoE-A2.7B | 53.8 / 10.2 | 17.4 / 8.9 | 70.0 / 9.2 | 46.3 / 11.3 | 42.7 / 9.4 | 35.5 / 6.7 | 54.9 / 8.0 |
| DeepSeek-LV2-A2.4B | 70.4 / 11.9 | 23.1 / 11.4 | 69.2 / 6.4 | 50.0 / 14.5 | 48.3 / 13.5 | 49.4 / 6.4 | 53.4 / 4.7 |
| DeepSeek-Coder-LV2-A2.4B | 85.7 / 14.1 | 56.4 / 11.7 | 69.5 / 15.2 | 72.6 / 13.9 | 64.9 / 13.6 | 63.8 / 5.2 | 55.9 / 13.3 |
| Qwen3-Coder-A3B | 86.4 / 10.3 | 81.2 / 8.1 | 90.7 / 11.1 | 92.7 / 12.4 | 72.9 / 11.0 | 87.5 / 8.1 | 77.5 / 9.8 |
| Qwen3-A3B | 90.0 / 11.3 | 90.7 / 9.8 | 93.3 / 12.1 | 92.7 / 9.9 | 74.9 / 7.8 | 90.5 / 8.8 | 81.6 / 10.5 |
| Qwen3-Next | 93.6 / 2.8 | 92.0 / 2.2 | 92.5 / 2.6 | 94.5 / 2.4 | 80.8 / 2.0 | 93.3 / 1.5 | 84.7 / 1.9 |
| GPT-OSS-A3.6B | 87.9 / 31.4 | 74.2 / 26.7 | 88.3 / 33.2 | 84.7 / 39.6 | 70.5 / 37.8 | 80.0 / 29.4 | 80.1 / 28.5 |
| DeepSeek-V2-A21B | 91.2 / 5.9 | 43.5 / 6.9 | 90.8 / 5.3 | 76.8 / 9.8 | 64.3 / 8.9 | 80.7 / 4.3 | 75.4 / 4.9 |
| DeepSeek-Coder-V2-A21B | 95.0 / 10.2 | 67.2 / 8.0 | 91.1 / 2.8 | 82.9 / 9.2 | 70.0 / 8.7 | 84.5 / 4.0 | 75.5 / 2.7 |
| DeepSeek-V2.5-A21B | 91.4 / 8.4 | 64.5 / 7.8 | 88.4 / 5.7 | 84.8 / 11.5 | 67.1 / 10.8 | 85.6 / 4.2 | 75.2 / 5.0 |
| Qwen3-A22B | 91.4 / 16.9 | 89.2 / 16.4 | 89.3 / 15.6 | 87.8 / 17.4 | 82.2 / 17.1 | 79.8 / 15.5 | 83.1 / 14.3 |
| GPT-OSS-A5.1B | 85.7 / 24.2 | 75.9 / 21.5 | 88.9 / 25.2 | 81.1 / 34.2 | 70.1 / 33.4 | 78.0 / 22.7 | 84.5 / 22.8 |

Table 6: Performance (%) and corresponding task Expert proportion(%), with $\eta_{expert} = 0.6$.

| Model | GSM8K (Math & Reasoning) | MATH (Math & Reasoning) | ARC$_c$ (Math & Reasoning) | HumanEval (Code) | MBPP (Code) | BBH (General) | MMLU (General) |
|---|---|---|---|---|---|---|---|
| DeepSeek-MoE-A2.8 | 59.6 / 8.9 | 13.2 / 18.3 | 52.4 / 21.9 | 46.9 / 6.8 | 47.3 / 9.5 | 42.3 / 18.9 | 44.8 / 31.9 |
| Qwen1.5-MoE-A2.7B | 53.8 / 11.2 | 17.4 / 15.0 | 70.0 / 21.8 | 46.3 / 4.3 | 42.7 / 5.5 | 35.5 / 12.6 | 54.9 / 36.5 |
| DeepSeek-LV2-A2.4B | 70.4 / 17.9 | 23.1 / 27.1 | 69.2 / 42.8 | 50.0 / 6.9 | 48.3 / 11.0 | 49.4 / 24.0 | 53.4 / 64.3 |
| DeepSeek-Coder-LV2-A2.4B | 85.7 / 12.5 | 56.4 / 23.6 | 69.5 / 25.8 | 72.6 / 6.6 | 64.9 / 10.1 | 63.8 / 29.3 | 55.9 / 38.6 |
| Qwen3-Coder-A3B | 86.4 / 29.9 | 81.2 / 43.6 | 90.7 / 36.8 | 92.7 / 13.3 | 72.9 / 20.2 | 87.5 / 34.4 | 77.5 / 50.8 |
| Qwen3-A3B | 90.0 / 22.4 | 90.7 / 33.6 | 93.3 / 31.3 | 92.7 / 15.1 | 74.9 / 22.4 | 90.5 / 27.0 | 81.6 / 44.4 |
| Qwen3-Next | 93.6 / 35.4 | 92.0 / 50.4 | 92.5 / 40.4 | 94.5 / 20.3 | 80.8 / 28.6 | 93.3 / 41.5 | 84.7 / 55.4 |
| GPT-OSS-A3.6B | 87.9 / 3.1 | 74.2 / 4.7 | 88.3 / 5.8 | 84.7 / 1.5 | 70.5 / 2.1 | 80.0 / 4.4 | 80.1 / 9.2 |
| DeepSeek-V2-A21B | 91.2 / 22.1 | 43.5 / 37.2 | 90.8 / 39.0 | 76.8 / 9.8 | 64.3 / 16.0 | 80.7 / 24.8 | 75.4 / 57.2 |
| DeepSeek-Coder-V2-A21B | 95.0 / 14.8 | 67.2 / 32.5 | 91.1 / 53.6 | 82.9 / 10.5 | 70.0 / 15.8 | 84.5 / 26.7 | 75.5 / 65.7 |
| DeepSeek-V2.5-A21B | 91.4 / 19.2 | 64.5 / 29.4 | 88.4 / 34.4 | 84.8 / 7.8 | 67.1 / 12.1 | 85.6 / 24.3 | 75.2 / 52.1 |
| Qwen3-A22B | 91.4 / 19.6 | 89.2 / 28.3 | 89.3 / 30.1 | 87.8 / 10.0 | 82.2 / 16.1 | 79.8 / 24.2 | 83.1 / 43.7 |
| GPT-OSS-A5.1B | 85.7 / 10.5 | 75.9 / 15.3 | 88.9 / 16.4 | 81.1 / 3.6 | 70.1 / 5.7 | 78.0 / 12.7 | 84.5 / 27.4 |

Table 7: Performance (accuracy %) and corresponding task Expert MUI(%) with $\eta_{expert} = 0.6$.

| Model | GSM8K | MATH | ARC$_c$ | HumanEval | MBPP | BBH | MMLU |
|---|---|---|---|---|---|---|---|
| | (Math & Reasoning) | (Math & Reasoning) | (Math & Reasoning) | (Code) | (Code) | (General) | (General) |
| OLMoE-0.5T | 2.8 / 6.9 | 3.3 / 8.2 | 25.7 / 14.2 | 8.5 / 10.7 | 10.4 / 11.6 | 25.7 / 11.5 | 27.2 / 11.2 |
| OLMoE-1T | 3.6 / 7.2 | 3.2 / 6.0 | 33.6 / 16.9 | 6.1 / 11.6 | 10.8 / 11.0 | 25.3 / 10.6 | 32.2 / 12.6 |
| OLMoE-1.5T | 3.8 / 8.5 | 3.8 / 7.0 | 42.2 / 12.1 | 8.0 / 11.0 | 10.6 / 10.9 | 25.6 / 11.5 | 41.3 / 11.4 |
| OLMoE-2T | 4.4 / 6.8 | 4.1 / 8.6 | 47.7 / 16.2 | 8.0 / 12.4 | 12.8 / 14.7 | 27.7 / 10.0 | 42.1 / 12.6 |
| OLMoE-2.5T | 4.7 / 7.4 | 4.0 / 8.0 | 50.8 / 14.6 | 11.6 / 12.1 | 17.4 / 13.5 | 26.2 / 11.9 | 44.0 / 11.5 |
| OLMoE-3T | 6.4 / 9.3 | 4.2 / 9.5 | 53.8 / 16.6 | 8.0 / 13.6 | 16.4 / 13.4 | 29.5 / 11.8 | 46.3 / 13.6 |
| OLMoE-3.5T | 6.4 / 9.3 | 4.5 / 6.3 | 54.5 / 17.0 | 9.1 / 13.9 | 16.0 / 14.6 | 20.0 / 7.5 | 46.7 / 14.2 |
| OLMoE-4T | 5.0 / 9.1 | 4.6 / 8.4 | 57.6 / 17.0 | 11.0 / 11.6 | 20.8 / 11.3 | 30.7 / 11.8 | 47.5 / 12.3 |
| OLMoE-4.5T | 4.4 / 9.7 | 4.7 / 8.2 | 56.7 / 15.7 | 11.6 / 13.5 | 18.0 / 12.8 | 31.8 / 11.0 | 48.5 / 12.0 |
| OLMoE-5T | 6.0 / 9.5 | 4.9 / 7.8 | 60.5 / 16.2 | 14.6 / 13.2 | 23.8 / 14.2 | 30.2 / 10.4 | 50.1 / 13.4 |

Table 8: Performance (%) and corresponding task Expert proportion(%), with $\eta_{expert} = 0.6$.

| Model | GSM8K | MATH | ARC$_c$ | HumanEval | MBPP | BBH | MMLU |
|---|---|---|---|---|---|---|---|
| | (Math & Reasoning) | (Math & Reasoning) | (Math & Reasoning) | (Code) | (Code) | (General) | (General) |
| OLMoE-0.5T | 2.8 / 20.4 | 3.3 / 38.8 | 25.7 / 19.0 | 8.5 / 7.2 | 10.4 / 9.6 | 25.7 / 13.7 | 27.2 / 26.1 |
| OLMoE-1T | 3.6 / 24.5 | 3.2 / 42.1 | 33.6 / 25.3 | 6.1 / 7.0 | 10.8 / 10.8 | 25.3 / 14.5 | 32.2 / 27.8 |
| OLMoE-1.5T | 3.8 / 25.3 | 3.8 / 36.6 | 42.2 / 30.6 | 8.0 / 7.8 | 10.6 / 11.1 | 25.6 / 14.8 | 41.3 / 28.5 |
| OLMoE-2T | 4.4 / 21.7 | 4.1 / 36.2 | 47.7 / 26.8 | 8.0 / 7.3 | 12.8 / 9.1 | 27.7 / 17.5 | 42.1 / 29.1 |
| OLMoE-2.5T | 4.7 / 25.3 | 4.0 / 36.0 | 50.8 / 27.7 | 11.6 / 8.0 | 17.4 / 10.9 | 26.2 / 14.0 | 44.0 / 27.7 |
| OLMoE-3T | 6.4 / 24.8 | 4.2 / 31.8 | 53.8 / 25.2 | 8.0 / 6.7 | 16.4 / 10.0 | 29.5 / 15.4 | 46.3 / 28.7 |
| OLMoE-3.5T | 6.4 / 24.2 | 4.5 / 45.1 | 54.5 / 25.0 | 9.1 / 6.6 | 16.0 / 9.2 | 20.0 / 22.6 | 46.7 / 28.4 |
| OLMoE-4T | 5.0 / 23.9 | 4.6 / 35.6 | 57.6 / 25.7 | 11.0 / 8.2 | 20.8 / 11.8 | 30.7 / 14.9 | 47.5 / 28.1 |
| OLMoE-4.5T | 4.4 / 19.3 | 4.7 / 34.3 | 56.7 / 24.7 | 11.6 / 6.7 | 18.0 / 10.5 | 31.8 / 14.8 | 48.5 / 28.8 |
| OLMoE-5T | 6.0 / 19.6 | 4.9 / 36.8 | 60.5 / 24.3 | 14.6 / 7.6 | 23.8 / 10.3 | 30.2 / 15.1 | 50.1 / 29.0 |

Table 9: Performance (accuracy %) and corresponding task Expert MUI(%) with $\eta_{expert} = 0.6$.

## B.3 EXPERT DISTRIBUTION

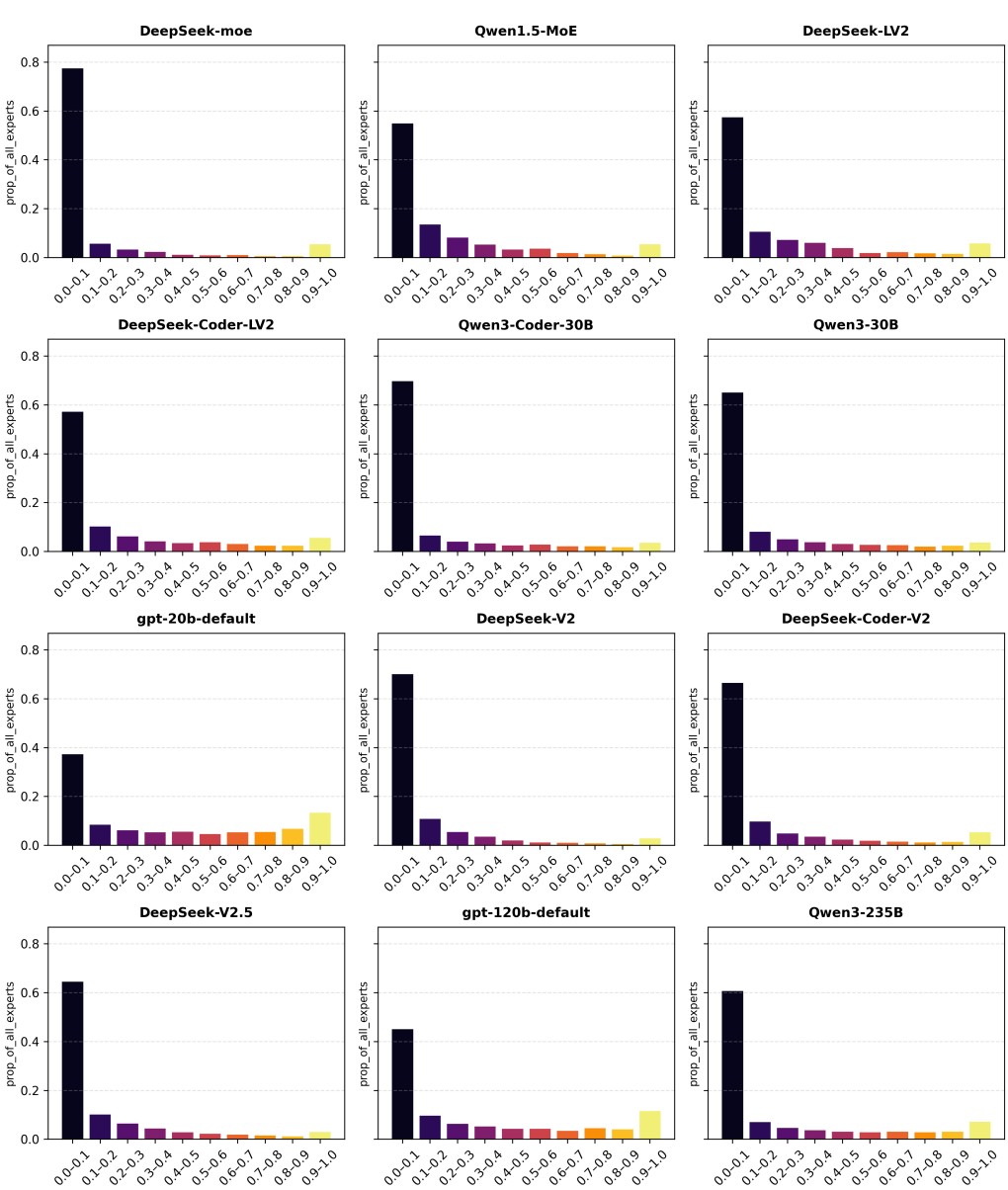

Figure 10: Frequency distribution of activated experts across all task instances for the selected models evaluated on the GSM8K benchmark.

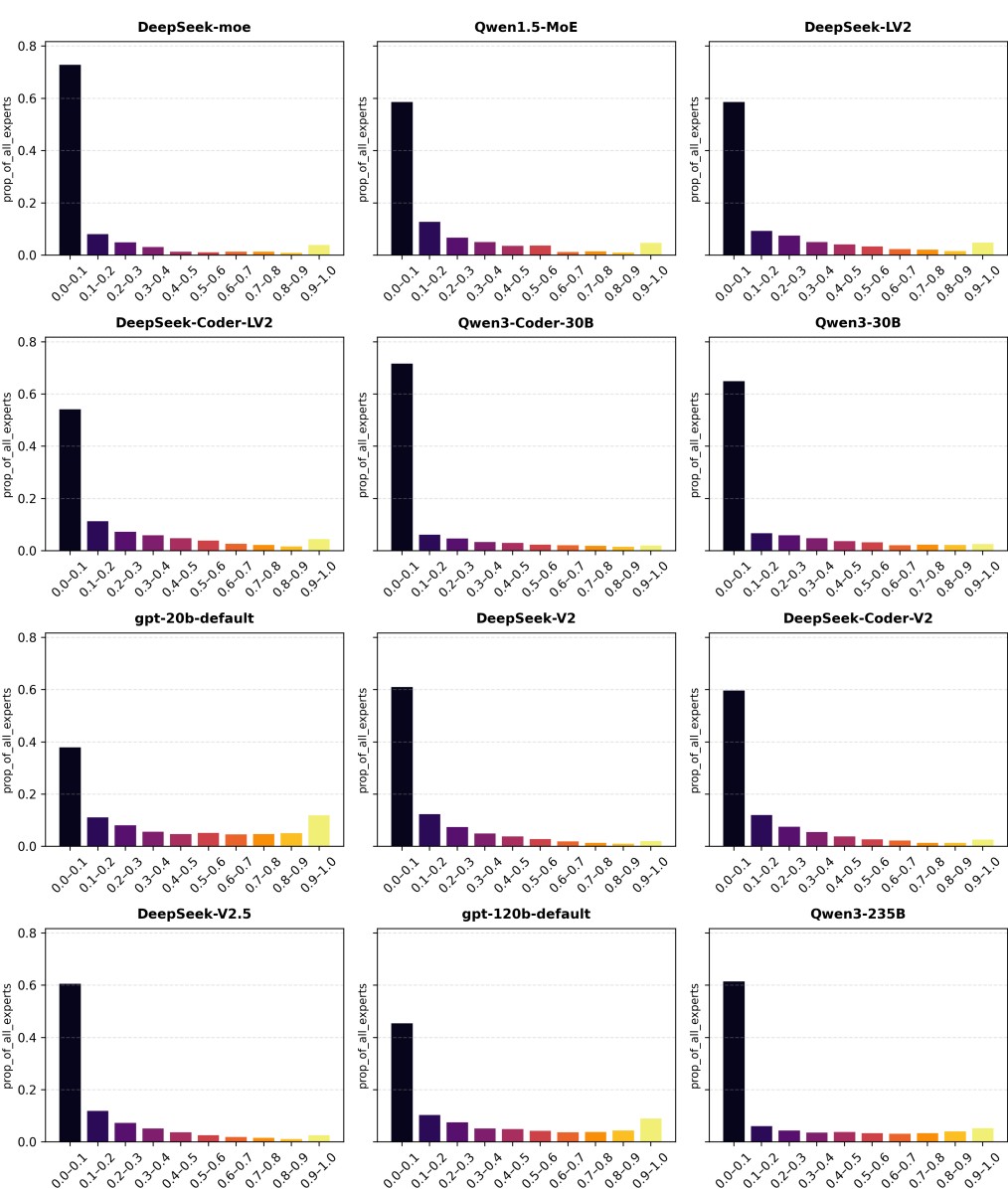

Figure 11: Frequency distribution of activated experts across all task instances for the selected models evaluated on the MATH benchmark.

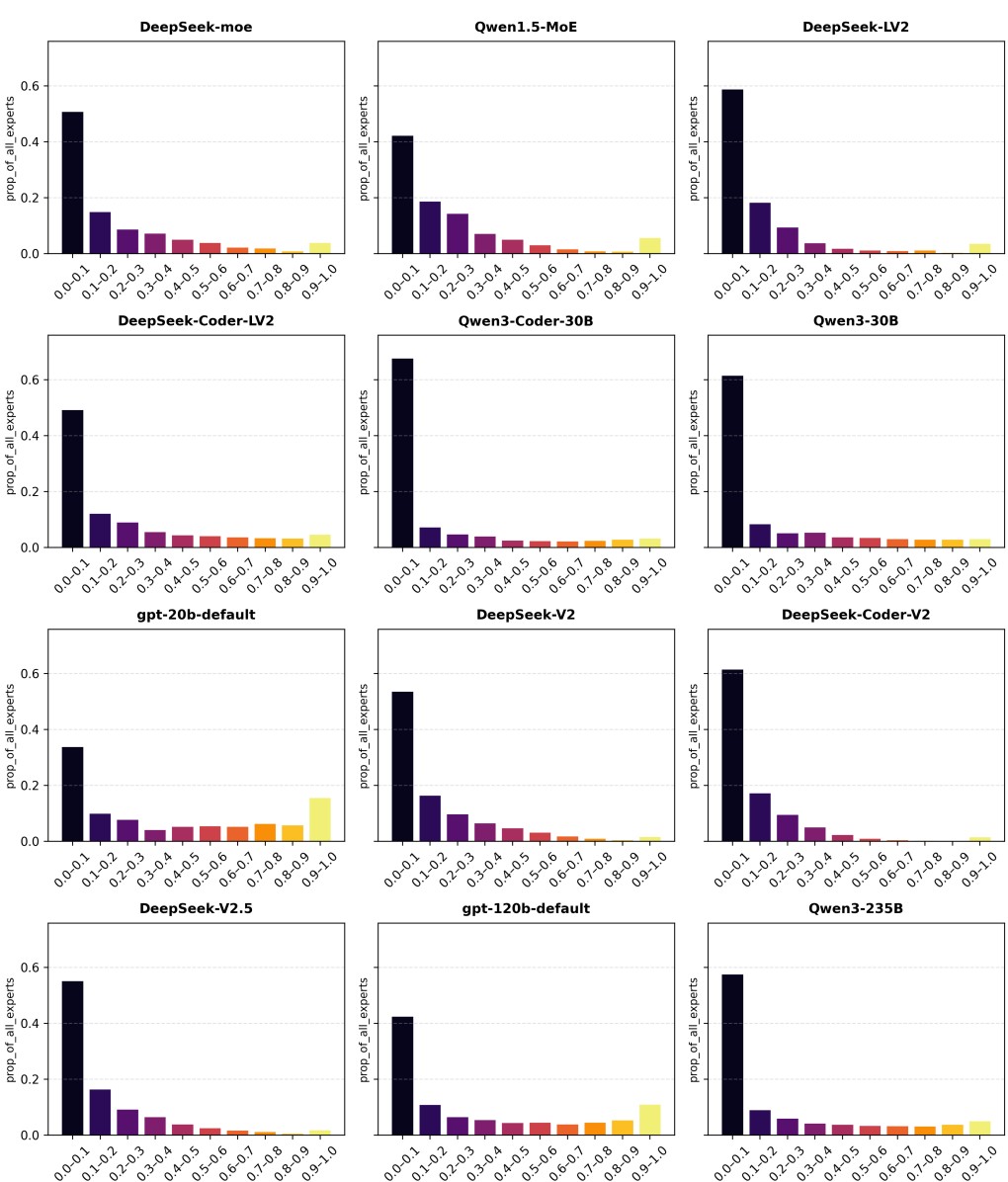

Figure 12: Frequency distribution of activated experts across all task instances for the selected models evaluated on the ARC benchmark.

Figure 13: Frequency distribution of activated experts across all task instances for the selected models evaluated on the HumanEval benchmark.

Figure 14: Frequency distribution of activated experts across all task instances for the selected models evaluated on the MBPP benchmark.

Figure 15: Frequency distribution of activated experts across all task instances for the selected models evaluated on the BBH benchmark.

Figure 16: Frequency distribution of activated experts across all task instances for the selected models evaluated on the MMLU benchmark.

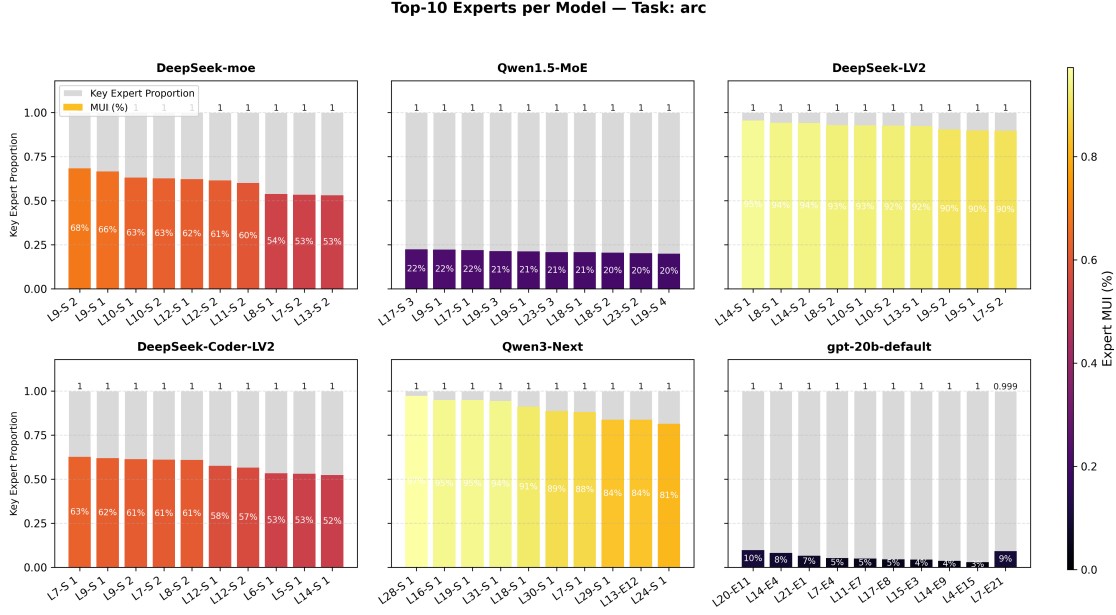

Figure 17: Top-10 experts (ranked by activation frequency Equation 5) for the selected MoE models with shared-expert structures (the exception GPT-OSS-20B model is included for comparison) on ARC. The corresponding MUI for each expert are also reported. Shared experts are denoted as $S_i$.

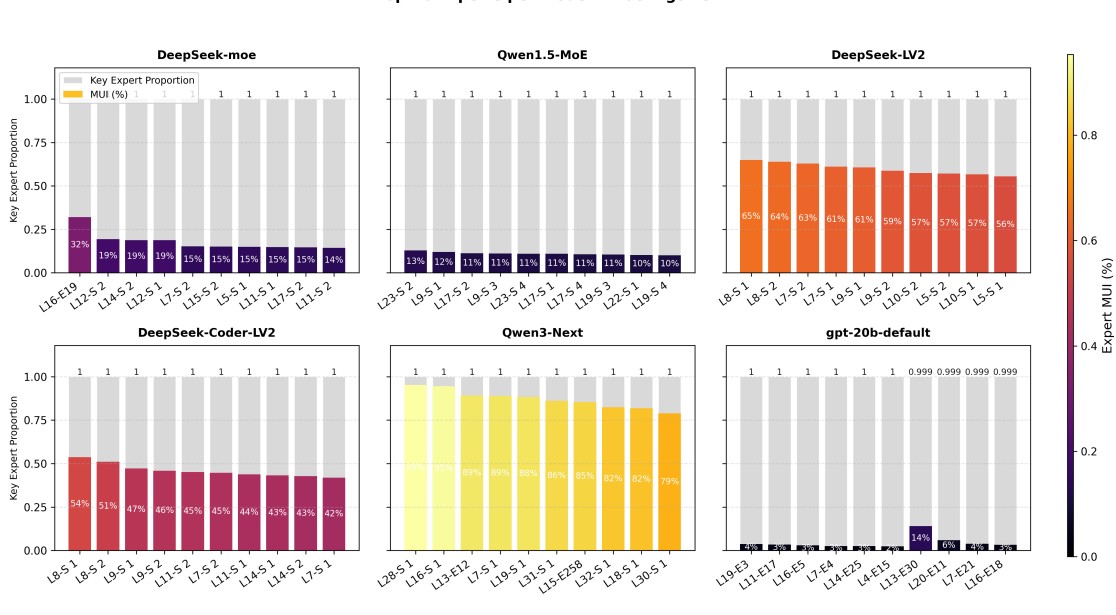

Figure 18: Top-10 experts (ranked by activation frequency Equation 5) for the selected MoE models with shared-expert structures (the exception GPT-OSS-20B model is included for comparison) on GSM8K. The corresponding MUI for each expert are also reported. Shared experts are denoted as $S_i$.

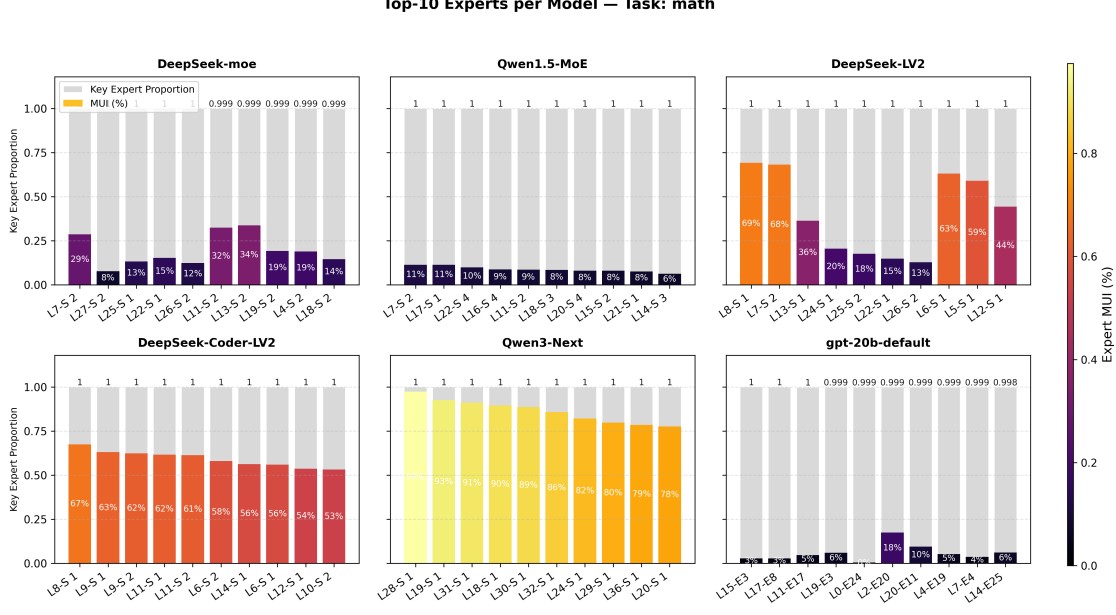

Figure 19: Top-10 experts (ranked by activation frequency Equation 5) for the selected MoE models with shared-expert structures (the exception GPT-OSS-20B model is included for comparison) on MATH. The corresponding MUI for each expert are also reported. Shared experts are denoted as $S_i$.

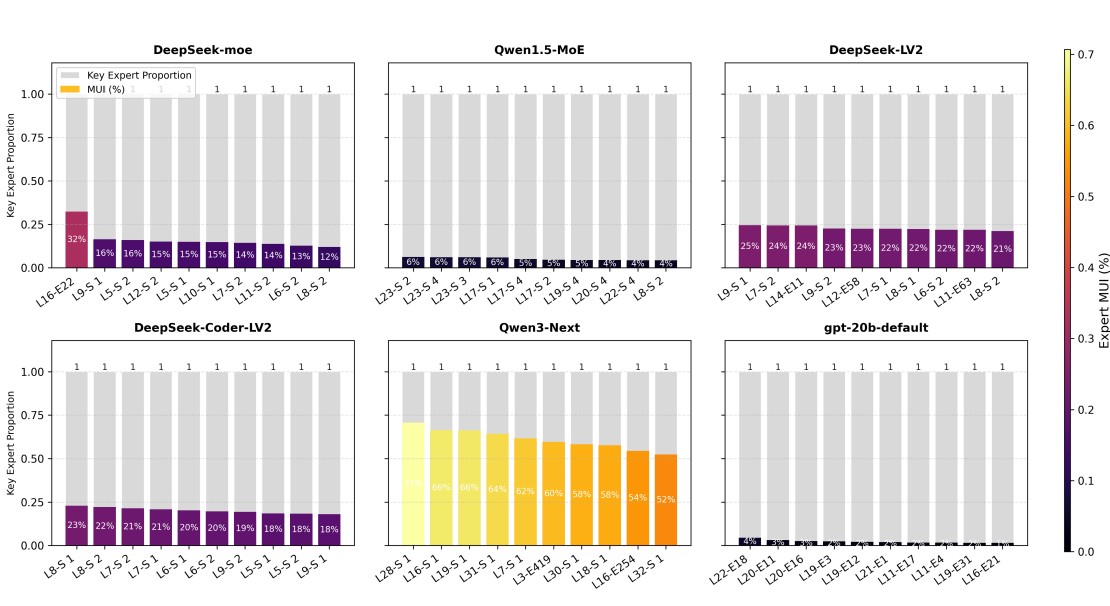

Figure 20: Top-10 experts (ranked by activation frequency Equation 5) for the selected MoE models with shared-expert structures (the exception GPT-OSS-20B model is included for comparison) on HumanEval. The corresponding MUI for each expert are also reported. Shared experts are denoted as $S_i$.

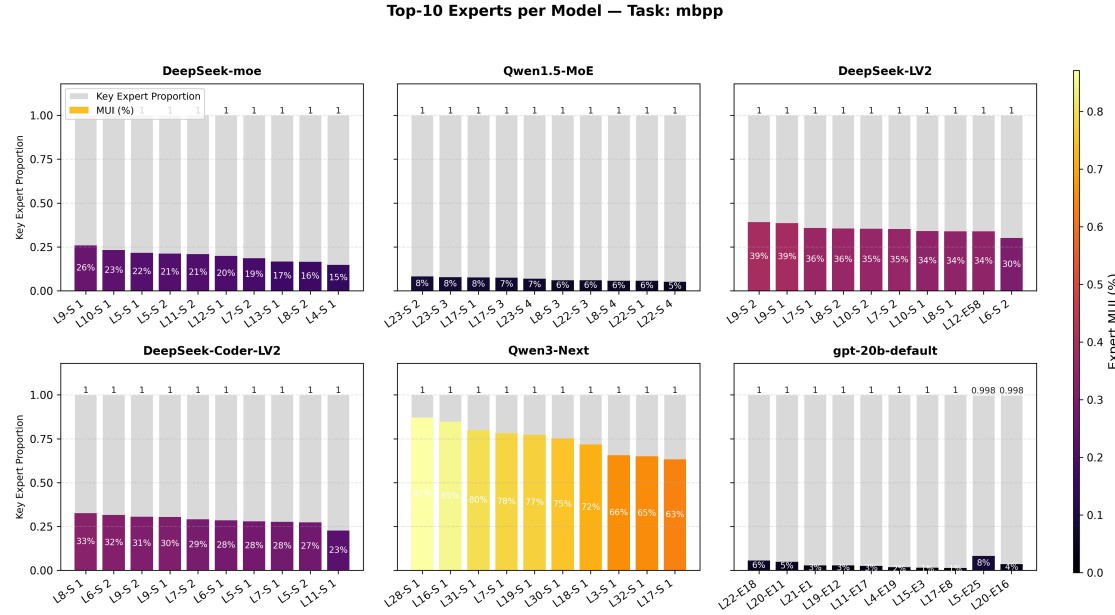

Figure 21: Top-10 experts (ranked by activation frequency Equation 5) for the selected MoE models with shared-expert structures (the exception GPT-OSS-20B model is included for comparison) on MBPP. The corresponding MUI for each expert are also reported. Shared experts are denoted as $S_i$.

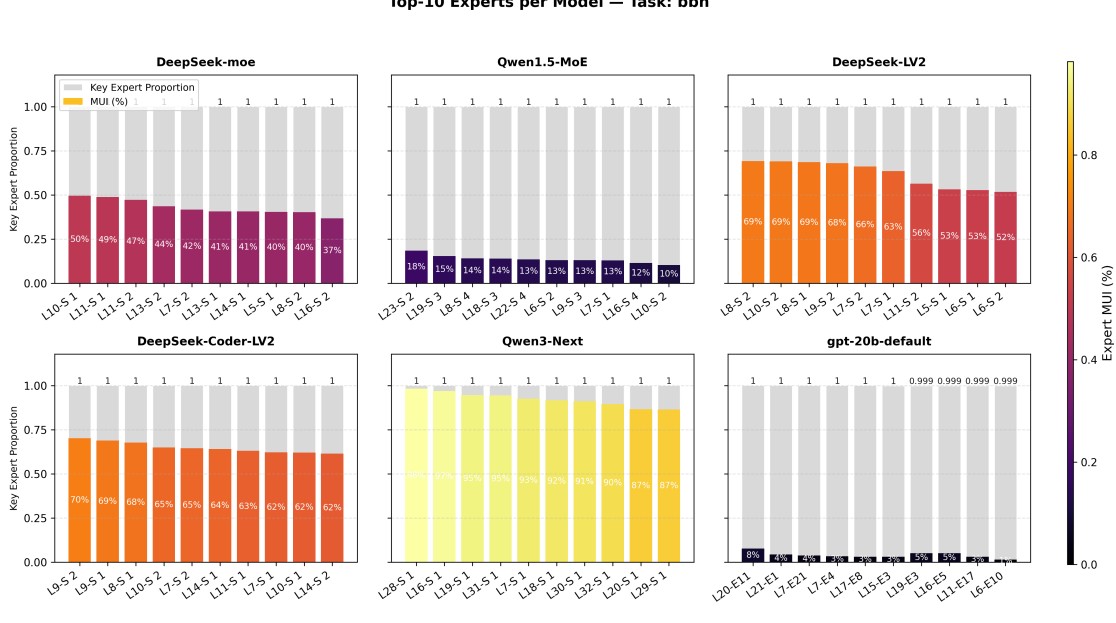

Figure 22: Top-10 experts (ranked by activation frequency Equation 5) for the selected MoE models with shared-expert structures (the exception GPT-OSS-20B model is included for comparison) on BBH. The corresponding MUI for each expert are also reported. Shared experts are denoted as $S_i$.

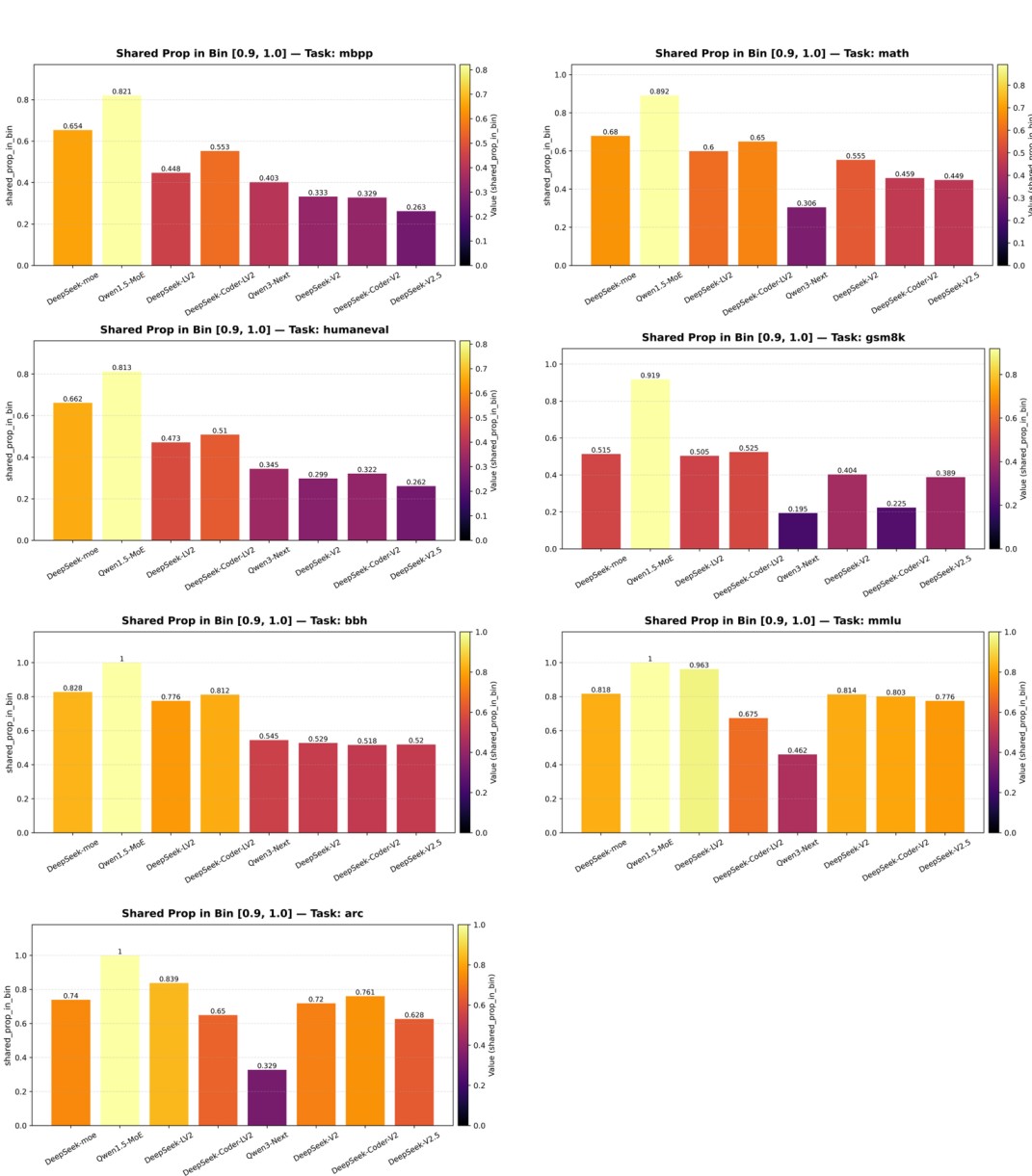

Figure 23: Proportion of shared experts among task experts with occurrence frequency greater than 0.9, showing a notably high level of overlap.

### B.4 MUI AND ACTIVATED EXPERT PROPORTION FOR DATA MEASUREMENT

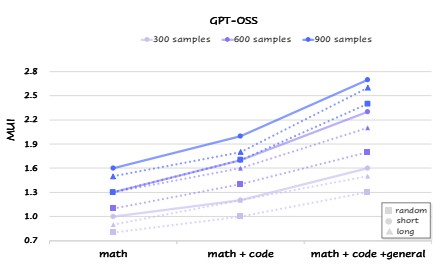

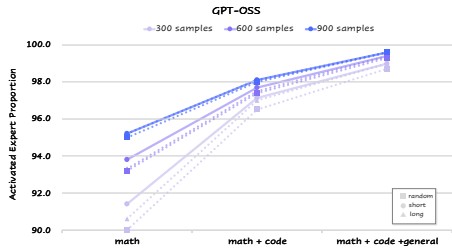

Figure 24: MUI across different data diversity.

Figure 25: Proportion of key experts across different data diversity .

## C ABLATION STUDY

In the main experiments, we primarily adopt a threshold of 0.1% for identifying key neurons. To test the robustness of this choice, we additionally evaluate smaller thresholds of 0.08% and 0.2% when computing MUI. The results, shown in Table 10 and Table 11, indicate extremely high consistency with the 0.1% setting: Pearson correlation = 0.9958 and Spearman correlation = 0.9965. These results demonstrate that our findings are stable across threshold variations.

| Model | GSM8K (Math & Reasoning) | MATH (Math & Reasoning) | ARC$_c$ (Math & Reasoning) | HumanEval (Code) | MBPP (Code) | BBH (General) | MMLU (General) |
|---|---|---|---|---|---|---|---|
| DeepSeek-MoE-A2.8 | 59.6 / 1.4 | 13.2 / 3.2 | 52.4 / 5.9 | 46.9 / 1.1 | 47.3 / 1.7 | 42.3 / 3.0 | 44.8 / 12.6 |
| Qwen1.5-MoE-A2.7B | 53.8 / 4.2 | 17.4 / 5.9 | 70.0 / 8.4 | 46.3 / 1.1 | 42.7 / 1.8 | 35.5 / 4.7 | 54.9 / 19.4 |
| DeepSeek-V2L-A2.4B | 70.4 / 4.1 | 23.1 / 6.6 | 69.2 / 9.0 | 50.0 / 1.4 | 48.3 / 2.5 | 49.4 / 4.7 | 53.4 / 20.3 |
| DeepSeek-Coder-V2L-A2.4B | 85.7 / 3.6 | 56.4 / 7.2 | 69.5 / 7.3 | 72.6 / 1.5 | 64.9 / 2.8 | 63.8 / 5.1 | 55.9 / 16.2 |
| Qwen3-Coder-A3B | 86.4 / 6.5 | 81.2 / 9.2 | 90.7 / 9.5 | 92.7 / 2.8 | 72.9 / 4.9 | 87.5 / 9.5 | 77.5 / 21.4 |
| Qwen3-A3B | 90.0 / 6.2 | 90.7 / 10.1 | 93.3 / 10.5 | 92.7 / 2.9 | 74.9 / 4.9 | 90.5 / 9.1 | 81.6 / 23.4 |
| Qwen3-Next | 93.6 / 4.6 | 92.0 / 7.7 | 92.5 / 8.6 | 94.5 / 1.7 | 80.8 / 3.0 | 93.3 / 6.5 | 84.7 / 22.1 |
| GPT-OSS-A3.6B | 87.9 / 1.4 | 74.2 / 2.2 | 88.3 / 2.7 | 84.7 / 0.7 | 70.5 / 1.1 | 80.0 / 2.0 | 80.1 / 5.8 |
| DeepSeek-V2-A21B | 91.2 / 5.4 | 43.5 / 11.3 | 90.8 / 13.7 | 76.8 / 2.2 | 64.3 / 4.4 | 80.7 / 7.4 | 75.4 / 31.5 |
| DeepSeek-Coder-V2-A21B | 95.0 / 5.0 | 67.2 / 11.7 | 91.1 / 12.3 | 82.9 / 2.8 | 70.0 / 5.3 | 84.5 / 8.1 | 75.5 / 26.2 |
| DeepSeek-V2.5-A21B | 91.4 / 5.6 | 64.5 / 10.4 | 88.4 / 13.2 | 84.8 / 2.1 | 67.1 / 4.1 | 85.6 / 7.6 | 75.2 / 30.0 |
| Qwen3-A22B | 91.4 / 5.4 | 89.2 / 7.8 | 89.3 / 9.7 | 87.8 / 2.2 | 82.2 / 3.9 | 79.8 / 6.8 | 83.1 / 21.4 |
| GPT-OSS-A5.1B | 85.7 / 3.7 | 75.9 / 5.6 | 88.9 / 6.7 | 81.1 / 1.4 | 70.1 / 2.3 | 78.0 / 5.0 | 84.5 / 14.8 |

Table 10: Performance and MUI, as determined by Equation 2 with threshold top $k = 0.08\%$.

| Model | GSM8K (Math & Reasoning) | MATH (Math & Reasoning) | ARC$_c$ (Math & Reasoning) | HumanEval (Code) | MBPP (Code) | BBH (General) | MMLU (General) |
|---|---|---|---|---|---|---|---|
| DeepSeek-MoE-A2.8 | 59.6 / 2.7 | 13.2 / 5.8 | 52.4 / 11.1 | 46.9 / 2.1 | 47.3 / 3.3 | 42.3 / 6.0 | 44.8 / 22.4 |
| Qwen1.5-MoE-A2.7B | 53.8 / 9.7 | 17.4 / 12.7 | 70.0 / 18.6 | 46.3 / 2.8 | 42.7 / 4.4 | 35.5 / 10.9 | 54.9 / 37.9 |
| DeepSeek-V2L-A2.4B | 70.4 / 7.6 | 23.1 / 11.8 | 69.2 / 16.3 | 50.0 / 3.0 | 48.3 / 5.0 | 49.4 / 9.0 | 53.4 / 33.1 |
| DeepSeek-Coder-V2L-A2.4B | 85.7 / 6.5 | 56.4 / 12.7 | 69.5 / 13.1 | 72.6 / 3.2 | 64.9 / 5.7 | 63.8 / 9.4 | 55.9 / 26.9 |
| Qwen3-Coder-A3B | 86.4 / 11.5 | 81.2 / 14.8 | 90.7 / 15.9 | 92.7 / 5.5 | 72.9 / 9.0 | 87.5 / 16.7 | 77.5 / 32.1 |
| Qwen3-A3B | 90.0 / 11.4 | 90.7 / 16.8 | 93.3 / 18.1 | 92.7 / 5.7 | 74.9 / 9.2 | 90.5 / 16.2 | 81.6 / 36.1 |
| Qwen3-Next | 93.6 / 9.7 | 92.0 / 15.0 | 92.5 / 16.7 | 94.5 / 3.7 | 80.8 / 6.5 | 93.3 / 13.3 | 84.7 / 38.3 |
| GPT-OSS-A3.6B | 87.9 / 2.9 | 74.2 / 4.1 | 88.3 / 5.8 | 84.7 / 1.4 | 70.5 / 2.2 | 80.0 / 4.3 | 80.1 / 11.6 |
| DeepSeek-V2-A21B | 91.2 / 10.2 | 43.5 / 20.2 | 90.8 / 24.4 | 76.8 / 4.7 | 64.3 / 8.8 | 80.7 / 14.0 | 75.4 / 48.0 |
| DeepSeek-Coder-V2-A21B | 95.0 / 9.7 | 67.2 / 21.1 | 91.1 / 22.5 | 82.9 / 5.9 | 70.0 / 10.7 | 84.5 / 15.2 | 75.5 / 42.3 |
| DeepSeek-V2.5-A21B | 91.4 / 11.0 | 64.5 / 19.0 | 88.4 / 23.8 | 84.8 / 4.6 | 67.1 / 8.5 | 85.6 / 14.2 | 75.2 / 46.4 |
| Qwen3-A22B | 91.4 / 9.9 | 89.2 / 13.5 | 89.3 / 17.1 | 87.8 / 4.5 | 82.2 / 7.5 | 79.8 / 12.4 | 83.1 / 33.7 |
| GPT-OSS-A5.1B | 85.7 / 7.6 | 75.9 / 10.8 | 88.9 / 13.2 | 81.1 / 3.0 | 70.1 / 4.9 | 78.0 / 10.1 | 84.5 / 26.5 |

Table 11: Performance and MUI, as determined by Equation 2 with threshold top $k = 0.2\%$.

In addition, we also experimented with alternative methods for computing neuron importance. Specifically, besides the default projection-based method, we tested using raw activations and projecting the entire upsampled outputs (details are provided in the Appendix A.4). We further conducted threshold ablations (Figure 27 and Figure 28) under these alternative formulations. After

evaluating different settings, we selected a threshold of 0.1% as the most appropriate and carried out experiments accordingly. For efficiency reasons, we restricted this analysis to four representative benchmarks: ARC, GSM8K, MBPP, and BBH. The resulting MUI values are shown in Table 12 and Table 13. We further compute similarity with Table 4, obtaining the following results: Cosine similarity: 0.9665, Pearson correlation: 0.8511, Spearman correlation: 0.7442; Cosine similarity: 0.9835, Pearson correlation: 0.9317, Spearman correlation: 0.9236. These results demonstrate that the overall trends remain highly consistent across different methods. Furthermore, within the same model scale, the ranking of models is stable, with GPT-OSS consistently exhibiting the lowest MUI.

| Model | GSM8K | $ARC_c$ | MBPP | BBH |
|---|---|---|---|---|
| DeepSeek-LV2-A2.4B | 70.4 / 3.4 | 69.2 / 8.1 | 50.0 / 2.3 | 49.4 / 4.4 |
| DeepSeek-Coder-LV2-A2.4B | 85.7 / 2.9 | 69.5 / 7.8 | 72.6 / 2.4 | 63.8 / 4.9 |
| Qwen3-A3B | 90.0 / 5.6 | 93.3 / 10.5 | 92.7 / 4.5 | 90.5 / 8.9 |
| GPT-OSS-A3.6B | 87.9 / 4.7 | 88.3 / 8.7 | 84.7 / 3.6 | 80.0 / 6.7 |
| DeepSeek-V2-A21B | 91.2 / 5.1 | 90.8 / 14.4 | 64.3 / 4.6 | 80.7 / 7.9 |
| DeepSeek-Coder-V2-A21B | 95.0 / 4.3 | 91.1 / 12.3 | 70.0 / 5.7 | 84.5 / 8.5 |

Table 12: Performance and MUI, as determined by Equation 9 with threshold top $k = 0.1\%$.

| Model | GSM8K | $ARC_c$ | MBPP | BBH |
|---|---|---|---|---|
| DeepSeek-LV2-A2.4B | 70.4 / 6.3 | 69.2 / 16.0 | 50.0 / 3.7 | 49.4 / 9.4 |
| DeepSeek-Coder-LV2-A2.4B | 85.7 / 5.4 | 69.5 / 12.0 | 72.6 / 4.5 | 63.8 / 9.7 |
| Qwen3-A3B | 90.0 / 8.3 | 93.3 / 14.0 | 92.7 / 6.3 | 90.5 / 12.3 |
| GPT-OSS-A3.6B | 87.9 / 1.4 | 88.3 / 3.1 | 84.7 / 1.3 | 80.0 / 2.2 |
| DeepSeek-V2-A21B | 91.2 / 6.3 | 90.8 / 16.1 | 64.3 / 4.8 | 80.7 / 9.3 |
| DeepSeek-Coder-V2-A21B | 95.0 / 5.0 | 91.1 / 14.2 | 70.0 / 5.4 | 84.5 / 9.4 |

Table 13: Performance and MUI as determined by Equation 10 with threshold top $k = 0.1\%$.

On the other hand, we also perform expert-level analyses using the same threshold of $\eta_{expert} = 0.6$ to define key experts. We compute results based on Equation 9 and Equation 10, reported in Table 14 and Table 15, and then compare them with the main results (Table 6). The similarities are as follows: Cosine similarity: 0.9947, Pearson correlation: 0.9863, Spearman correlation: 0.9533; Cosine similarity: 0.9794, Pearson correlation: 0.9503, Spearman correlation: 0.9030. These results demonstrate that expert-level measurements yield highly consistent trends and results. Moreover, GPT consistently exhibits the highest proportion of key experts, further confirming our findings.

| Model | GSM8K | $ARC_c$ | MBPP | BBH |
|---|---|---|---|---|
| DeepSeek-LV2-A2.4B | 70.4 / 9.6 | 69.2 / 6.8 | 50.0 / 11.7 | 49.4 / 6.2 |
| DeepSeek-Coder-LV2-A2.4B | 85.7 / 12.7 | 69.5 / 13.0 | 72.6 / 10.9 | 63.8 / 6.0 |
| Qwen3-A3B | 90.0 / 10.2 | 93.3 / 10.8 | 92.7 / 7.4 | 90.5 / 6.6 |
| GPT-OSS-A3.6B | 87.9 / 30.1 | 88.3 / 36.5 | 84.7 / 40.9 | 80.0 / 28.9 |
| DeepSeek-V2-A21B | 91.2 / 7.1 | 90.8 / 5.9 | 64.3 / 10.0 | 80.7 / 4.7 |
| DeepSeek-Coder-V2-A21B | 95.0 / 12.8 | 91.1 / 3.3 | 70.0 / 10.3 | 84.5 / 5.2 |

Table 14: Performance and corresponding task Expert proportion (the neuron is finding using Equation 9), with $\eta_{expert} = 0.6$.

| Model | GSM8K | $ARC_c$ | MBPP | BBH |
|---|---|---|---|---|
| DeepSeek-LV2-A2.4B | 70.4 / 16.8 | 69.2 / 9.0 | 50.0 / 20.6 | 49.4 / 14.0 |
| DeepSeek-Coder-LV2-A2.4B | 85.7 / 23.7 | 69.5 / 23.0 | 72.6 / 23.7 | 63.8 / 13.6 |
| Qwen3-A3B | 90.0 / 12.0 | 93.3 / 12.4 | 92.7 / 8.4 | 90.5 / 8.9 |
| GPT-OSS-A3.6B | 87.9 / 33.1 | 88.3 / 35.9 | 84.7 / 40.5 | 80.0 / 29.0 |
| DeepSeek-V2-A21B | 91.2 / 8.3 | 90.8 / 8.2 | 64.3 / 11.7 | 80.7 / 5.9 |
| DeepSeek-Coder-V2-A21B | 95.0 / 13.5 | 91.1 / 3.3 | 70.0 / 11.6 | 84.5 / 6.1 |

Table 15: Performance and corresponding task Expert proportion (the neuron is finding using Equation 10), with $\eta_{expert} = 0.6$.

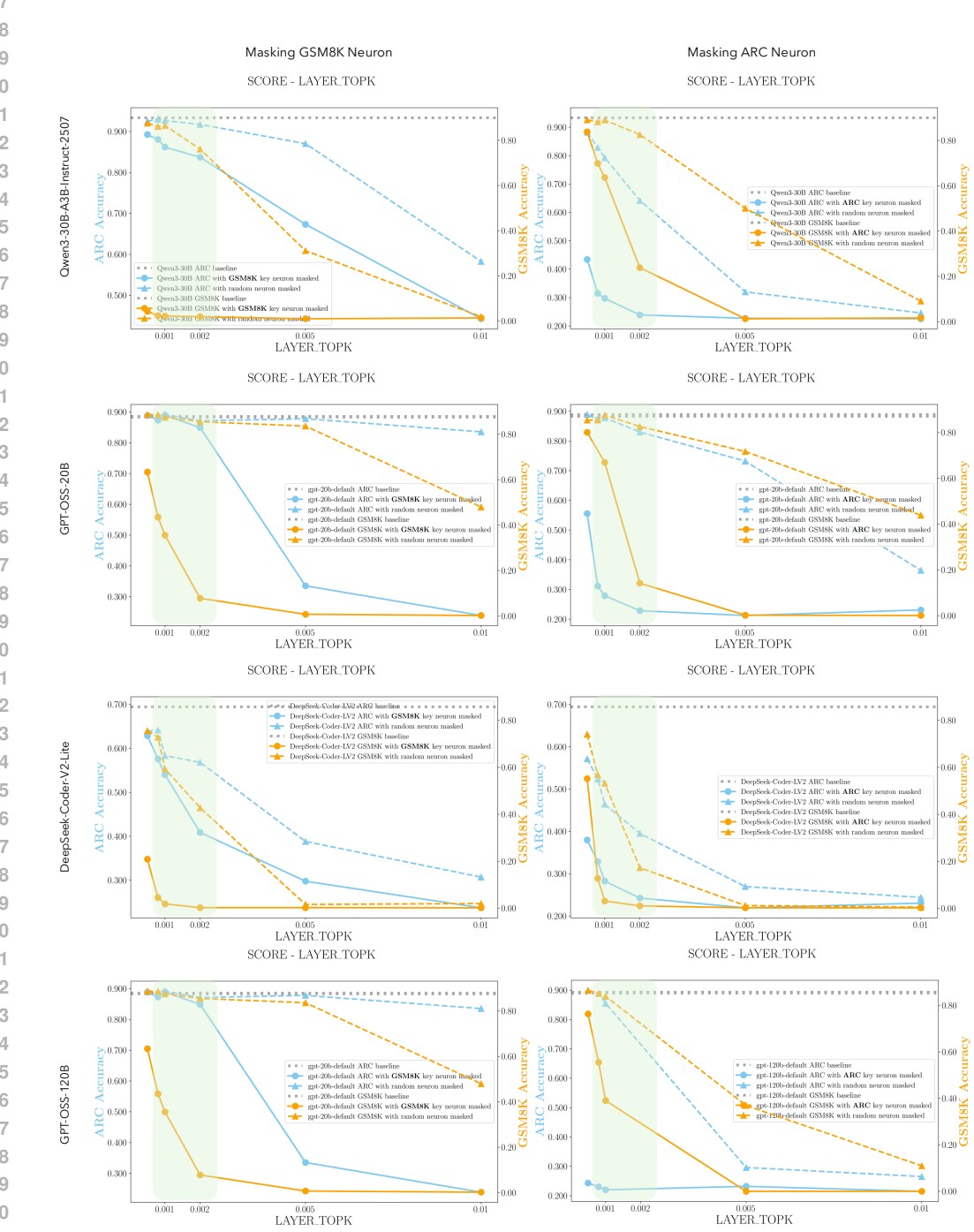

Figure 26: Performance accuracy (ACC) of the on the ARC and GSM8K datasets, with key neurons masked specifically for the **ARC** dataset or the **GSM8K** dataset. Key neurons are identified using Equation 2 and a pre-defined threshold function (Detailed in Appendix A.3). The threshold value used for our MUI analysis —0.1% to 0.2%, is visually indicated by a green box . The performance impact of masking an equivalent number of key neurons as in the ARC / GSM8K dataset on the corresponding model is represented with a dashed line.

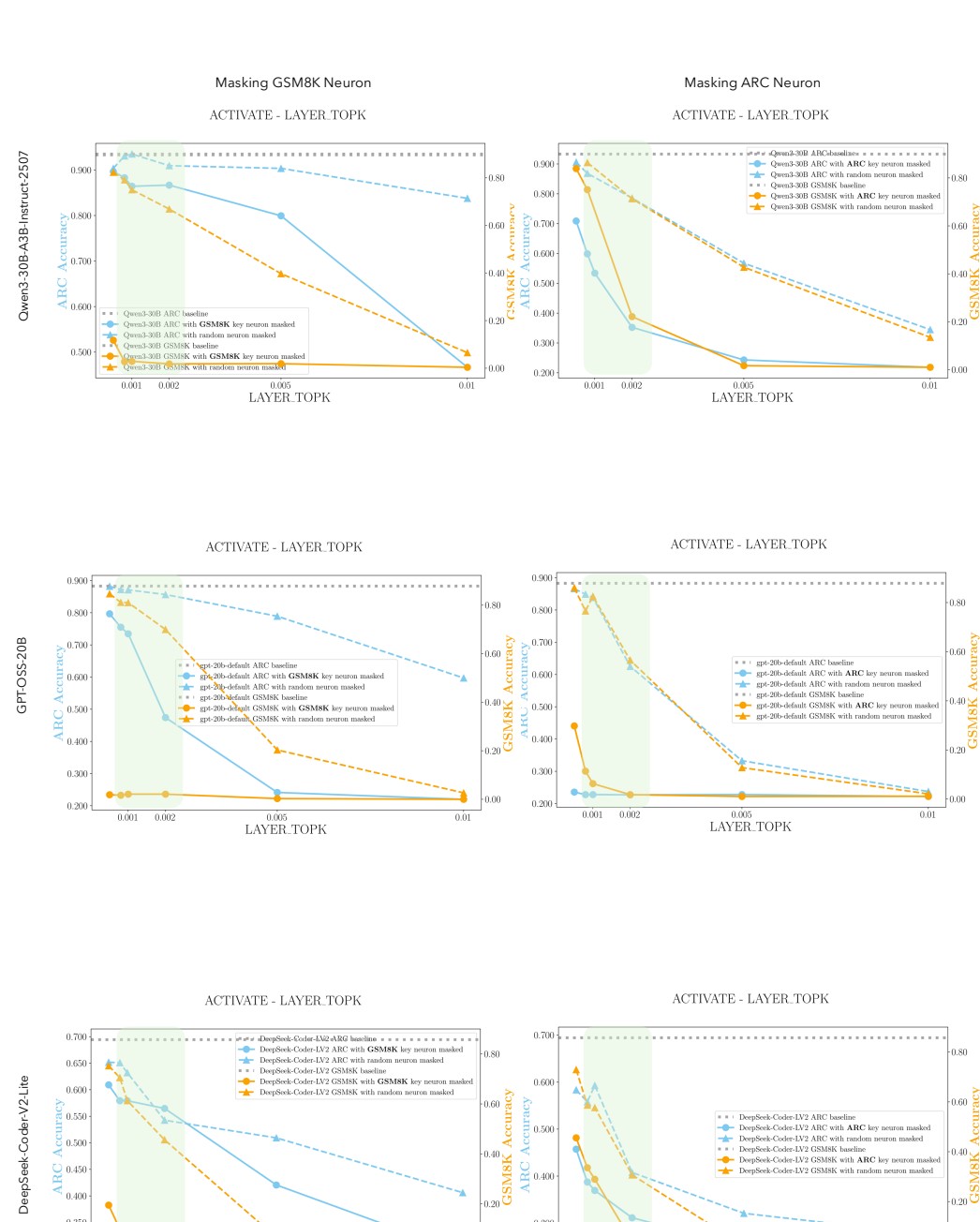

Figure 27: Performance accuracy (ACC) of the on the ARC and GSM8K datasets, with key neurons masked specifically for the **ARC** dataset or the **GSM8K** dataset. Key neurons are identified using Equation 9 and a pre-defined threshold function (Detailed in Appendix A.3). The threshold value used for our MUI analysis —0.1% to 0.2%, is visually indicated by a  green box . The performance impact of masking an equivalent number of key neurons as in the ARC / GSM8K dataset on the corresponding model is represented with a dashed line.

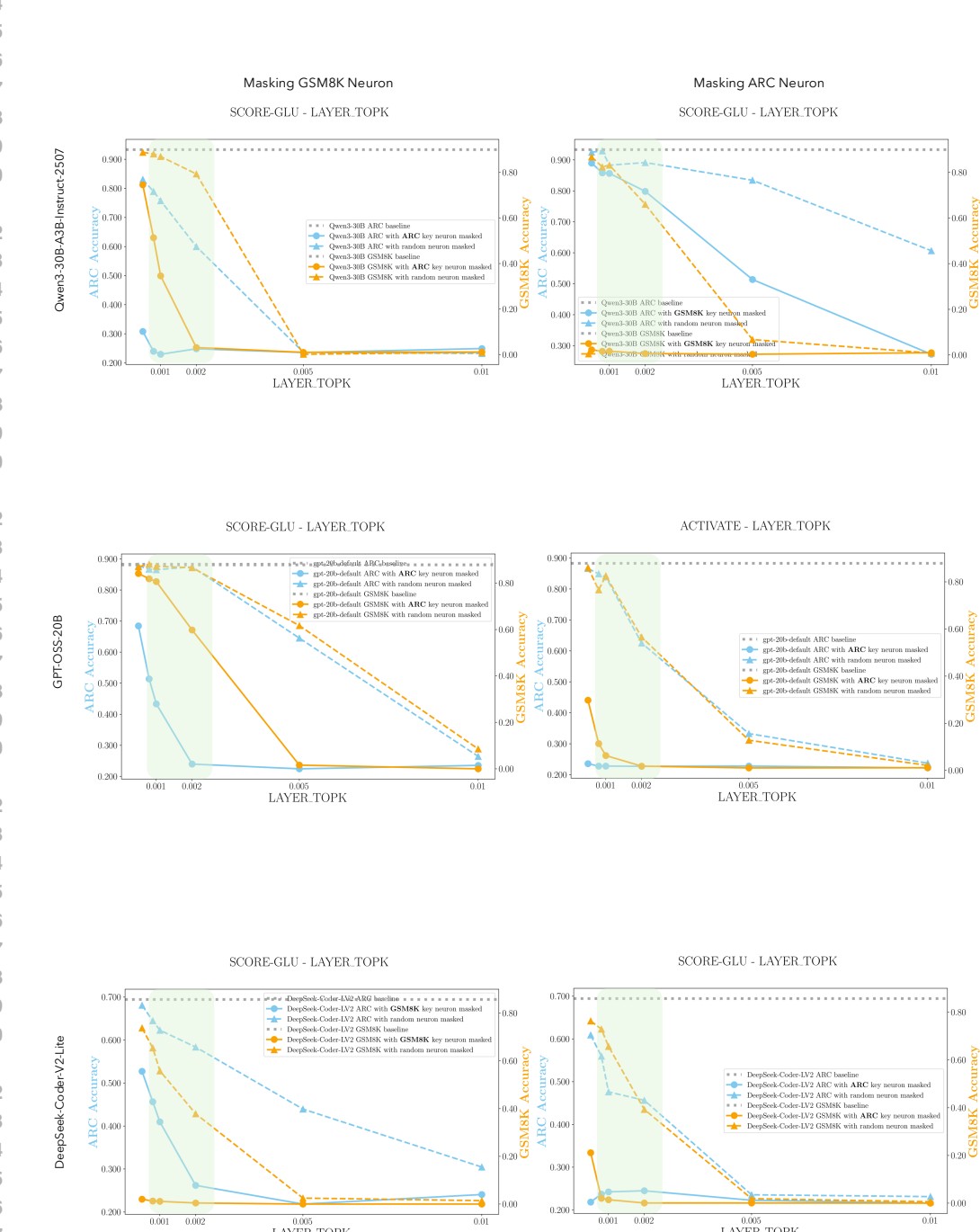

Figure 28: Performance accuracy (ACC) of the on the ARC and GSM8K datasets, with key neurons masked specifically for the **ARC** dataset or the **GSM8K** dataset. Key neurons are identified using Equation 10 and a pre-defined threshold function (Detailed in Appendix A.3). The threshold value used for our MUI analysis —0.1% to 0.2%, is visually indicated by a green box . The performance impact of masking an equivalent number of key neurons as in the ARC / GSM8K dataset on the corresponding model is represented with a dashed line.

