# OpenReview forum: "Beyond Benchmarks: Understanding Mixture-of-Experts Models through Internal Mechanisms"
_ICLR.cc/2026/Conference — ICLR 2026 Conference Withdrawn Submission_

### Official Review · Reviewer_fTfs · 2025-10-28

**Soundness:** 2
**Presentation:** 3
**Contribution:** 2
**Rating:** 6
**Confidence:** 2

**Summary:**

The paper extends the MUI (Model Utilization Index) introduced by Cao et al. (Model Utility Law: Evaluating LLMs beyond Performance through Mechanism Interpretable Metric, https://arxiv.org/abs/2504.07440) to MoE models. Based on the original MUI formulation as well as the extension, the authors perform a series of experiments on open-weights MoE models and families: GPT-OSS, Qwen, DeepSeek, and OLMoE. The main results include (1) tracking the neuron utilization as model family evolves, (2) tracking the changes to MUI during training based on OLMoE checkpoints (3) investigation of the expert collaboration and neuron activation within experts, and (4) investigation of the MoE .
It is shown with ablations that some aspects of the work that could seem arbitrary ($\eta$'s, MUI equation) are actually meaningful.

**Strengths:**

1. Results are based on investigation across different model families.
2. The work is based on an established method, contributing to the soundness of the method and simplifying comparison with dense models.
3. Presentation quality is good and the paper is easy to read.

**Weaknesses:**

1. The training dynamics experiments are limited to OLMoE, although at least one other open-weights family with intermediate checkpoints exists (Xue et al., Openmoe: an early effort on open mixture-of-experts language models)
2. I think the paper would benefit from (some amount of) comparative analysis with dense models. The baseline method (Cao et al. https://arxiv.org/abs/2504.07440) is proposed for dense models, however little comparison of this paper's results for MoE models with the results for dense models is made.
3. No custom-trained models. Although such addition is not necessary and would require considerable compute, measuring MUI and related metrics on models which differ only in key aspects (such as whether there are shared experts) while the rest of the pipeline (dataset, etc.) is kept the same between models would allow for much more authoritative claims.
4. Figure 3 would benefit from having a separate version for each benchmark, containing all intermediate checkpoints (even if just in the appendix)
5. As far as I can understand, the method used can only be used to investigate model behavior in a statistical, aggregated manner (across a dataset).

**Questions:**

1. The authors consider an expert important based on the contribution of its neurons, however another natural candidate for this task emerges - the routing decisions (at least in case of non-shared experts). Have the authors considered comparing their method (in sections related to expert-leven contribution) to an approach based on the expert-use frequency?
2. In figure 8. the MUI rises steadily as the number of samples is increased. Does that happen indefinitely? Is it possible that higher MUI is simply correlated with the dataset being larger?
3. Is there any more evidence for there existing separate ``accumulating'' and ``evolving'' phases? Does something interesting happen during these phases? These are introduced, however it seems their importance is very limited, as they do not even transfer between tasks.
4. Can the method be used to interpret the model behavior in a more fine-grained manner, i.e., within a single sample, not across a dataset.
5. Can the ablations (section 4) be performed on experts instead of neurons? Considering the paper introduces the notion of key experts, such ablation could prove that such notion is indeed meaningful.

---

> ### Author Response · Authors · 2025-11-19
> **Response to Reviewer fTfs (part 1)**
>
> Thanks for your insightful suggestions and views. Here is our response:
>
> ### **General Response**
>
> Thank you for recognizing that this work explores a novel and promising direction. Measuring the generalization or capability of MoE models is an inherently challenging task—especially as model ability improves with scale. Beyond the risk of in-domain contamination in public benchmarks, it is fundamentally difficult to use finite test data to faithfully reflect a model’s unbounded generalization space. Nevertheless, given the importance of model evaluation, we believe it is worthwhile to explore alternative perspectives beyond performance metrics. Our work takes a first step by analyzing MoE models through their internal mechanisms, leveraging multiple interpretability techniques to understand how neuron/expert utilization relate to evaluation outcomes. As with any new approach, we acknowledge that controversy and limitations are inevitable. **We kindly ask the reviewers to consider whether, based on our detailed analyses, our reasoning is internally consistent and whether this line of inquiry opens up meaningful and promising research directions**—even if it is not possible to exhaustively address all aspects within a single paper.
>
> Our central claim—lower MUI and better performance indicates better generalization—is supported from multiple perspectives:
>
> 1. **Model analysis within the same series:** When analyzing models with clearly different intrinsic capacities within the same family, we observe that stronger MoE models consistently achieve higher benchmark performance while exhibiting lower MUI values. In particular, DeepSeek-Coder-V2-Lite is further pre-trained from a checkpoint of DeepSeek-V2-Lite, which provides a clean comparison: the performance gap between the two, combined with the widespread deployment and adoption of the former, as suggested by its higher download counts, offers a practical and training-driven demonstration of this trend.
> 2. **Supporting evidence from prior work:** Similar patterns have been observed in controlled and dense-model settings. Anthropic (2023) [1] showed that as a single ReLU network improves its generalization, its parameters become increasingly specialized. Cao et al. (2025) [2] reported consistent findings for dense models.
>
> Taken together, these findings suggest that MUI serves as an indicator of intrinsic model capacity and generalization.
>
> [1] Superposition, Memorization, and Double Descent
>
> [2] Model Utility Law: Evaluating LLMs beyond Performance through Mechanism Interpretable Metric

---

> ### Author Response · Authors · 2025-11-19
> **Response to Reviewer fTfs (part 2)**
>
> #### Regarding the concerns "The training dynamics experiments are limited to OLMoE"
>
> - **Response 4.1:** Thanks for your suggestion. The main reason we do not include additional models in the training-dynamics analysis is that very few fully open-source MoE models release intermediate training checkpoints.  Although OpenMoE is available, it is not a fully MoE architecture—only certain intermediate layers use MoE architecture.
>
> #### Regarding the suggestion "benefit from (some amount of) comparative analysis with dense models."
>
> - **Response 4.2:** Thanks for your suggestion.  Since MUI is designed to reflect intrinsic MoE behaviors, directly comparing MUI values between MoE and dense transformers is not mature yet, given their fundamentally different structures and activation patterns. We plan to investigate this more in future work.
>
> #### Regarding the suggestion of addition custom-trained models
>
> - **Response 4.3:**  Thanks for your suggestion. We have already begun related training experiments. But given the scope of this work, we do not elaborate on them here.
>
> #### Regarding the suggestion "Figure 3 would benefit from having a separate version for each benchmark"
>
> - **Response 4.4:** Thank you for the suggestion. Given the amount of experimental content and analysis already included in this work, we are unable to include a per-benchmark version of Figure 3 in the current main text. However, we will incorporate this improvement in the camera-ready version.
>
> #### Regarding the reason why "the method  investigate model behavior dataset" than data sample.
>
> - **Response 4.5:** Our method uses MUI to analyze how neuron/expert utilization relates to evaluation outcomes, which naturally conducting on at the dataset level to analysis together with the corresponding performance. Although the interpretability methods underlying MUI can operate at the per-sample level.
>
> #### Regarding the expert important based on the contribution of its neurons how about the the routing decisions
>
> - **Response 4.6:** We assume the reviewer is asking whether we consider the routed decisions for experts important. Since Equation (2) computes the importance score by incorporating the router’s decisions, the analysis of the neuron’s located expert importance naturally already incorporates the routing behavior. If the reviewer has further questions, we are welcome to discuss more.
>
> #### Regarding the understanding of MUI &  number of samples
>
> - **Response 4.7:**  The change in MUI with the number of samples is due to different samples test different capabilities. In essence, increasing the number of diverse samples increases coverage of the model’s behaviors, which naturally leads to higher MUI. **Simply repeating the same samples does not increase MUI.**
>
>   Because MUI is defined as a percentage, adding more distinct evaluation samples causes MUI to increase gradually. In the idealized case, MUI may approach 100%.
>
> #### Regarding the question "Is there any more evidence and something interesting happen during these phases "
>
> - **Response 4.8:** Additional evidence comes from two directions.
>
>   First, dense models also show similar trends in MUI during training, where MUI changes in a way that reflects shifts in the model’s learning dynamics. Second, although the setting is very different, a related phenomenon appears in [1]: during the training of a single ReLU network, the authors observed two distinct phases in the loss curve. The network first undergoes a memorization-like phase (loss decreases), then a period where the loss increases and decreases again, during which the parameters become more specialized and generalization improves.
>
>   We believe that exploring these two phases further could help us better understand how models memorize, how they learn, and how generalization emerges. We are currently conducting follow-up investigations in this direction. As stated in our General Response, we also hope reviewers will consider whether our work opens up meaningful and promising research directions—even if not all aspects can be exhaustively addressed in a single paper.
>
> #### Regarding the question "Can the ablations (section 4) be performed on experts instead of neurons"
>
> - **Response 4.9:**  The ablations in Section 4 are designed to validate the neuron selection. Since the importance of a neuron naturally reflects the importance of the expert it belongs to, masking an entire expert would have an impact no smaller than masking the key neurons within that expert.

---

> ### Author Response · Authors · 2025-11-24
> **Dear Reviewer fTfs**
>
> Dear Reviewer fTfs,
>
> Thank you once again for your review and comments. We kindly request your feedback and comments, especially regarding whether our response addressed your concerns. Your feedback is crucial to us. We apologize for any inconvenience caused.

---

> ### Author Response · Authors · 2025-11-28
> **Response to Reviewer fTfs**
>
> Dear Reviewer,
>
> Thank you again for your review. **As the rebuttal/discussion period is drawing to a close,** we respectfully request your attention to our response. We believe the points raised in our rebuttal effectively address your initial concerns. If you find that our responses have clarified your concerns, we would be grateful for any final comments or a re-evaluation of your score. We appreciate your valuable time and input.

---

### Official Review · Reviewer_W8fg · 2025-10-31

**Soundness:** 2
**Presentation:** 2
**Contribution:** 2
**Rating:** 4
**Confidence:** 5

**Summary:**

- Demonstrates that evolved models achieve better performance with lower neuron utilization, suggesting MUI reflects generalization capability beyond benchmark scores. Training exhibits a two-phase trajectory: an "Accumulating" phase where MUI increases with performance, followed by an "Evolving" phase where MUI decreases as the model becomes more efficient.

- Reveals that task completion emerges from collaborative contributions of multiple experts rather than isolated specialization. Stronger models exhibit higher proportions of key experts working together, with GPT-OSS showing the highest collaboration rates. However, in architectures with shared experts, task responsibility concentrates heavily within the shared pool, potentially limiting the diversity benefits of distributed experts.

**Strengths:**

- Introduces a principled neuron-level analysis methodology for MoE models that explicitly accounts for routing mechanisms and expert-level behaviors, extending beyond performance-only evaluation. The approach is validated through multiple ablation studies (threshold variations, alternative importance methods) and causal intervention experiments (neuron masking) showing high correlation across formulations.

- Analyzes 13 production MoE models (20B-200B parameters) plus 10 training checkpoints across 7 diverse benchmarks, revealing consistent patterns across model families. The discovery that evolved models achieve lower MUI while improving performance provides actionable insights for model development.

- Provides the first evidence that shared experts dominate task responsibility, challenging assumptions about expert specialization in MoE architectures. The two-phase training trajectory discovery (Accumulating → Evolving) offers practical guidance for monitoring training and identifying when models need more task-specific data.

**Weaknesses:**

- While the paper claims to investigate "internal mechanisms," it primarily provides correlational observations without explaining why lower MUI corresponds to better generalization or what semantic features the utilized neurons represent. The neuron contribution method relies on logit lens projections, which have known limitations—they assume linear representation geometry and may not capture the full computational role of neurons in multi-layer nonlinear transformations. The paper never validates whether the identified "key neurons" actually encode interpretable features (e.g., arithmetic operations, syntactic patterns) or are merely correlates of the output distribution. Without grounding MUI in actual semantic or functional properties of neurons, it remains unclear whether low MUI reflects genuine capability compression or simply different architectural routing patterns.

- The paper demonstrates that MUI changes during training but provides no concrete methodology for using MUI to improve model development. The recommendation that "OLMoE needs more coding data" is post-hoc and could have been identified through standard per-task performance monitoring. Critically missing: (1) Can MUI predict downstream performance on held-out tasks? (2) What MUI threshold indicates a model is ready to transition training phases? The "Takeaway" boxes offer vague guidance ("monitoring performance alone is insufficient") without quantitative decision rules or experimental validation that MUI-guided interventions actually improve final model quality.

- The neuron masking experiments only validate that identified neurons are causally relevant but do not establish that MUI uniquely captures generalization over alternative explanations. Lower MUI in evolved models could reflect: (1) better optimization leading to sparser representations, (2) architectural differences, (3) post-training procedures, or (4) differences in training data scale/quality. The correlation between low MUI and strong performance in GPT-OSS is particularly questionable since these models differ in every dimension (architecture, scale, data, training recipe) from the comparison models.

- The definition of "key neurons" depends critically on the top-0.1% threshold, yet the justification is primarily empirical convenience rather than principled theory.

**Questions:**

- Can you provide evidence that MUI captures generalizable capacity rather than architectural artifacts? Please report MUI for models trained on identical datasets with only architectural variations (e.g., different expert counts, with/without shared experts), or show that MUI computed on training data predicts held-out task performance.

- What are the concrete, quantitative decision rules for using MUI in practice? Can you specify: (1) MUI thresholds that indicate when to transition from "Accumulating" to "Evolving" phase training, (2) what MUI delta triggers data mixture adjustments, and (3) empirical validation that MUI-guided interventions (e.g., adding coding data when coding MUI stays high) actually improve final model quality compared to standard training? Without this, MUI remains a post-hoc diagnostic rather than an actionable training tool.

- How do you disentangle neuron function from routing artifacts in your MUI measurements? Given that your method projects through the unembedding matrix (Equation 2), have you verified that "key neurons" encode interpretable features relevant to the task?

---

> ### Author Response · Authors · 2025-11-19
> **Response to Reviewer W8fg (part 1)**
>
> Thanks for your review and suggestion. It appears there may have been some misunderstandings regarding the contributions of our work, and some inquiries were not clearly articulated. Based on our interpretation, we have endeavored to address these questions as comprehensively as possible. We are open to and look forward to further discussion to clarify any concerns.
>
> ### **General Response**
>
> Thank you for recognizing that this work explores a novel and promising direction. Measuring the generalization or capability of MoE models is an inherently challenging task—especially as model ability improves with scale. Beyond the risk of in-domain contamination in public benchmarks, it is fundamentally difficult to use finite test data to faithfully reflect a model’s unbounded generalization space. Nevertheless, given the importance of model evaluation, we believe it is worthwhile to explore alternative perspectives beyond performance metrics. Our work takes a first step by analyzing MoE models through their internal mechanisms, leveraging multiple interpretability techniques to understand how neuron/expert utilization relate to evaluation outcomes. As with any new approach, we acknowledge that controversy and limitations are inevitable. **We kindly ask the reviewers to consider whether, based on our detailed analyses, our reasoning is internally consistent and whether this line of inquiry opens up meaningful and promising research directions**—even if it is not possible to exhaustively address all aspects within a single paper.
>
> Our central claim—lower MUI and better performance indicates better generalization—is supported from multiple perspectives:
>
> 1. **Model analysis within the same series:** When analyzing models with clearly different intrinsic capacities within the same family, we observe that stronger MoE models consistently achieve higher benchmark performance while exhibiting lower MUI values. In particular, DeepSeek-Coder-V2-Lite is further pre-trained from a checkpoint of DeepSeek-V2-Lite, which provides a clean comparison: the performance gap between the two, combined with the widespread deployment and adoption of the former, as suggested by its higher download counts, offers a practical and training-driven demonstration of this trend.
> 2. **Supporting evidence from prior work:** Similar patterns have been observed in controlled and dense-model settings.  [1] showed that as a single ReLU network improves its generalization, its parameters become increasingly specialized.  [2] reported consistent findings for dense models.
>
> Taken together, these findings suggest that MUI serves as an indicator of intrinsic model capacity and generalization.
>
> [1] Superposition, Memorization, and Double Descent
>
> [2] Model Utility Law: Evaluating LLMs beyond Performance through Mechanism Interpretable Metric

---

> ### Author Response · Authors · 2025-11-19
> **Response to Reviewer W8fg (part 2)**
>
> #### Regarding the question "without explaining why lower MUI corresponds to better generalization" and  "MUI captures generalizable capacity rather than architectural artifacts?"
>
> - **Response 3.1**: As a metric reflecting intrinsic MoE behaviors, MoE-MUI (%) is not yet mature for broad cross-model comparisons (e.g., across very different architectures, sizes). This is precisely why our core conclusions are drawn within the same model series, using models of similar size and, in some cases like, identical architectures. Please refer to the General Response for a detailed explanation of why lower MUI correlates with stronger generalization.
>
>   At the same time, we apply several mitigation strategies to further alleviate the architectural influence, using top-k% rather than absolute top-k scoring.
>
> #### Regarding the misunderstanding of our work "relies on logit lens projections" and "should validates whether the identified "key neurons" actually encode interpretable features"
>
> - **Response 3.2**: Our work **does not** rely on any single interpretability method. Interpretability remains a rapidly evolving field, and there is currently no consensus on how to definitively identify which neurons encode a "feature." Different interpretability techniques define and detect "key neurons" in different ways. For this reason, our work intentionally leverages multiple perspectives (Section 2.2 for the main setting, with additional implementations in Section 4 and Appendix C) when analyzing MoE models. Aiming to provide a diversity internal evaluation perspective on MoE models and to get consistent conclusion, **rather than to determine whether any method perfectly identifies feature-encoding neurons.**
>
>   Importantly, despite employing different interpretability method, the analyses in Section 4 and Appendix C show highly consistent trends, which further validates our findings.
>
> #### Regarding the misinterpretation that "no methodology for using MUI to improve model development" and "OLMoE needs more coding data could have been identified through standard per-task performance monitoring."
>
> - **Response 3.3:** Our analysis in Section 3.3 shows that lower MUI corresponds to better generalization (please refer to Response 3.1). This naturally suggests a direct training-time guideline: when analyzing a model during training, **we should not rely solely on monitoring performance** on a limited set of benchmarks. Instead, monitoring both performance and MUI provides a more informative picture of the model’s internal generalization behavior.
>
>   In addition, our checkpoint-based analysis of OLMoE reveals more fine-grained and actionable insights into training dynamics: 1.Accumulating Phase — performance ↑, MUI ↑, the model appears to recruit a broader set of neurons/experts to support learning. 2.Evolving Phase — performance ↑, MUI ↓, with continued training, the model reaches a stage where performance continues to improve while MUI decreases, aligning with the trend we identify as indicative of stronger generalization. This finer-grained distinction helps us better diagnose where the model stands during training.
>
>   Based on these observations, we propose a practical guideline for model development (Section 3.4): A well-trained MoE model should ideally exhibit both high performance and low MUI. When a model remains in the Accumulating Phase, appropriate training adjustments—such as providing additional or more diverse training data, applying model-specific training strategies, or continuing training—may help the model reducing MUI and improving generalization.
>
>   If this does not fully resolve your concern, **we would greatly appreciate clarification on what specific form of  "concrete methodology" or "quantitative decision rules" you are expecting.** We are open to discuss.
>
> #### Regarding the misunderstanding "Can MUI predict downstream performance on held-out tasks "
>
> - **Response 3.4:**  MoE-MUI is using to reflect intrinsic behaviors rather than predict the performance. Please refer to General Response for more detail.
>
> #### Regarding the concern "do not establish that MUI uniquely captures generalization"
>
> - **Response 3.5**: Please refer to Response 3.1 and General Response.

---

> ### Author Response · Authors · 2025-11-19
> **Response to Reviewer W8fg (part 3)**
>
> #### Regarding the concern "key neurons" depends critically on the top-0.1% threshold" and no "principled theory"
>
> - **Response 3.6:** As we known, there is currently no principled method for selecting neurons. In this work, our goal is therefore to leverage existing interpretability practices to analyze MoE models (Response 3.2). The use of a top-k threshold is a widely adopted practice in interpretability analyses [3, 4, 5]. Our choice of top-0.1% setting is empirically motivated, as detailed in Section 4. Moreover, to ensure robustness, we also conduct ablation study on the threshold in Appendix C, and the results show consistent trends, further supporting the stability of our findings.
>
>   If the you has recommendations for a more principled theoretical framework or alternative thresholding schemes that could strengthen our analysis, we would be very grateful for such guidance.
>
> #### Regarding the question "Please report MUI for models trained on identical datasets with only architectural variations"
>
> - **Response 3.7:** Please refer to Response3.1.
>
> #### Regarding the question "MUI thresholds that indicate when to transition from "Accumulating" to "Evolving" phase training"
>
> - **Response 3.8:**  Accumulating and Evolving phases are **not defined** by MUI value. Instead, they are characterized by the joint trend between model performance and MUI over training iterations. For a detailed explanation, please refer to Response 3.3.
>
> #### Regarding the question "what MUI delta triggers data mixture adjustments"
>
> - **Response 3.9:** Please refer to Response 3.3 for more detailed information.
>
> #### Regarding the concern "MUI remains a post-hoc diagnostic rather than an actionable training tool." and the suggestion no "improve final model quality compared to standard training"
>
> - **Response 3.10:** I assume that when you say "MUI remains a post-hoc diagnostic", you mean that MUI only evaluates intrinsic MoE behaviors. But isn’t that exactly the purpose of our work—to propose a new evaluation perspective for MoE models?
>
>   As for the phrase "rather than an actionable training tool". If you are referring to how MUI could guide training, please refer to Response 3.3
>
>   Regarding your suggestion "improve final model quality compared to standard training", we are unsure what you intend. Are you asking us to demonstrate that "lower MUI corresponds to better generalization"? If so, please see our Response 3.1.
>
>   We welcome you to clarify your question so that we can address the concern appropriately.
>
> #### Regarding the question "how disentangle neuron function from routing artifacts"
>
> - **Response 3.11:** We are not sure what you mean by "how disentangle neuron function from routing artifacts". Could you please clarify this in more detail?
>
>   Our definition of neuron importance is given in Section 2.2 and Appendix C. In addition, we conduct ablation experiments in Section 4 (Figure 26 to 28 for more detail),  demonstrate the validity of our neuron selection.
>
>
>
> [3] Neuron-Level Knowledge Attribution in Large Language Models, EMNLP
>
> [4] Finding and Editing Multi-Modal Neurons in Pre-Trained Transformers, ACL
>
> [5] Neuron Shapley: Discovering the Responsible Neurons, NeurIPS

---

> ### Author Response · Authors · 2025-11-24
> **Response to Reviewer W8fg**
>
> Dear Reviewer W8fg,
>
> Thank you once again for your review and comments. We kindly request your feedback and comments, especially regarding whether our response addressed your concerns. Your feedback is crucial to us. We apologize for any inconvenience caused.

---

> ### Author Response · Authors · 2025-11-28
> **Response to Reviewer W8fg**
>
> Dear Reviewer,
>
> Thank you again for your review.
> **As the rebuttal/discussion period is drawing to a close,** we respectfully request your attention to our response. We believe the points raised in our rebuttal effectively address your initial concerns.
> Specifically, regarding the scope of our contribution and the methodology, we clarified that our work focuses on MoE analysis using established techniques rather than defining a new interpretation method. We would also like to highlight that the Logit Lens approach continues to be a tool for analyzing internal representations in high-impact research published [1, 2, 3].
> We would also genuinely appreciate it if you could suggest specific novel methods that you believe would be more suitable for our analysis.
> If you find that our responses have clarified your concerns, we would be grateful for any final comments or a re-evaluation of your score.
> We appreciate your time and input.
>
>
>
> [1] Neuron-Level Knowledge Attribution in Large Language Models, EMNLP, 2024
>
> [2] Do Llamas Work in English? On the Latent Language of Multilingual Transformers, ACL, 2024
>
> [3] MoE Lens - An Expert Is All You Need, ICLR, 2025

---

### Official Review · Reviewer_SNTy · 2025-11-01

**Soundness:** 3
**Presentation:** 3
**Contribution:** 2
**Rating:** 6
**Confidence:** 4

**Summary:**

This paper presents a novel approach to understanding Mixture-of-Experts (MoE) models by analyzing their internal mechanisms rather than relying solely on benchmark performance. The authors adapt the Model Utilization Index (MUI) (orginally proposed by [Cao et al., 2025](https://arxiv.org/abs/2504.07440)) to MoE architectures and conduct extensive analyses on 3 different series (13 discrete models) of MoE models (20B-200B parameters) and several checkpoints OLMoE-7b .

The authors extend the MUI metric (originally for dense models) to MoE architectures by:

1. **Neuron-level Analysis**: Computing contribution scores for each neuron considering routing weights:

    $$
    f_{neuron}(i, j, l, \hat{y} | x) = G^l_i(x^l) \cdot (x^lW^l_{g,i}) \cdot W^l_{d,i}[j] \cdot W_{head}[:, \hat{y}]
    $$

    where $G^l_i$ are routing weights and $W$ matrices are projections in SwiGLU

2. **Model-level MUI**: Proportion of activated neurons relative to total:

    $$
    \text{MUI}(T) = \frac{|\bigcup N_{activated}(s_i)|}{N \times L \times (|E_s| + |E_r|)}
    $$

3. **Expert-level Analysis**: Identifying key experts with frequency threshold $\eta_{\text{expert}} = 0.6$  and computing expert-specific MUI

**Strengths:**

- Methodological contribution by extending the Model Utilization Index (MUI) from dense models to MoE architectures. This isn't a trivial extension - thoughtfully adapted the metric to account for MoE-specific characteristics like routing mechanisms and the distinction between shared and routed experts. The formulation in Equations 4-7 provides both model-level and expert-level granularity, enabling multi-scale analysis that wasn't previously available.
- The work offers practical value:
    - MUI as a training monitor can detect when specific capabilities need more data (e.g., OLMoE's persistent accumulating phase for coding)
    - The metric can potentially detect benchmark overfitting/leakage by comparing MUI trends with performance gains
    - Insights about shared experts inform architectural decisions for future MoE designs
- The paper tries addresses a critical issue that, why do some models (particularly GPT series) perform really well in real-world applications while not always dominating benchmarks? The discovery that GPT-OSS less neuron activity/utilization provides a mechanistic explanation for the same. This isn't just correlation - the causal masking experiments prove these neurons are actually doing the work. This suggests that **generalization quality can be measured by representational efficiency**, fundamentally changing how we should evaluate models beyond benchmark accuracy.
****

**Weaknesses:**

- It claims experts "collaborate" but doesn't analyze what individual experts learn:
    - Conducting semantic analysis of expert specialization could reveal whether experts specialize or remain largely redundant
    - Probing expert representations would help determine if experts learn complementary features that explain their collaborative behavior, providing mechanistic insight into why MoE architectures succeed
- The core claim that lower MUI indicates better generalization lacks justification. In Section 3.2, the authors state: "If we assume that these newer models indeed possess higher true capability and stronger generalization... then MUI may serve as an indicator." This is circular reasoning - they assume what they're trying to prove. The paper needs either:
    - Theoretical analysis connecting neuron sparsity to generalization bounds
    - Controlled experiments where generalization is measured independently (e.g., through distribution shift experiments, not just benchmark performance)
    - References to established theory linking sparsity and generalization in neural networks
- The neuron importance formula (Equation 2) multiplies by routing weights G_i(x), which are **already sparse by design** in MoE models. The paper then claims this shows "better utilization" but this seems to be just measuring the routing sparsity, not actual neuron utility

**Questions:**

- I would recommend that please look into Figure 1 make it more clear and easy to understand the illustration overall is not very clear
- Noticed inconsistent model naming like DeepSeek-V2-Light / DeepSeek-V2-Lite. Please follow the consistently and Lite is the right one to avoid any sort of confusion among readers. [very minor, but just spotting it out]

---

> ### Author Response · Authors · 2025-11-19
> **Response to Reviewer SNTy (part 1)**
>
> Thank you for your careful review and suggestions. Here is our response:
>
> ### **General Response**
>
> Thank you for recognizing that this work explores a novel and promising direction. Measuring the generalization or capability of MoE models is an inherently challenging task—especially as model ability improves with scale. Beyond the risk of in-domain contamination in public benchmarks, it is fundamentally difficult to use finite test data to faithfully reflect a model’s unbounded generalization space. Nevertheless, given the importance of model evaluation, we believe it is worthwhile to explore alternative perspectives beyond performance metrics. Our work takes a first step by analyzing MoE models through their internal mechanisms, leveraging multiple interpretability techniques to understand how neuron/expert utilization relate to evaluation outcomes. As with any new approach, we acknowledge that controversy and limitations are inevitable. **We kindly ask the reviewers to consider whether, based on our detailed analyses, our reasoning is internally consistent and whether this line of inquiry opens up meaningful and promising research directions**—even if it is not possible to exhaustively address all aspects within a single paper.
>
> Our central claim—lower MUI and better performance indicates better generalization—is supported from multiple perspectives:
>
> 1. **Model analysis within the same series:** When analyzing models with clearly different intrinsic capacities within the same family, we observe that stronger MoE models consistently achieve higher benchmark performance while exhibiting lower MUI values. In particular, DeepSeek-Coder-V2-Lite is further pre-trained from a checkpoint of DeepSeek-V2-Lite, which provides a clean comparison: the performance gap between the two, combined with the widespread deployment and adoption of the former, as suggested by its higher download counts, offers a practical and training-driven demonstration of this trend.
> 2. **Supporting evidence from prior work:** Similar patterns have been observed in controlled and dense-model settings. Anthropic (2023) [1] showed that as a single ReLU network improves its generalization, its parameters become increasingly specialized. Cao et al. (2025) [2] reported consistent findings for dense models.
>
> Taken together, these findings suggest that MUI serves as an indicator of intrinsic model capacity and generalization.
>
> [1] Superposition, Memorization, and Double Descent
>
> [2] Model Utility Law: Evaluating LLMs beyond Performance through Mechanism Interpretable Metric

---

> ### Author Response · Authors · 2025-11-19
> **Response to Reviewer SNTy (part 2)**
>
> #### Regarding the suggestion for "analyze what individual experts learn"
>
> - **Response 2.1:** The primary goal of this work is to analyze the internal mechanisms of MoE models to provide a new perspective for evaluating model capabilities. As a first attempt, we intentionally focus on aggregate neuron and expert activation patterns, which allow us to characterize model-level behaviors. Thus, we did not include detailed analyses of specific individual experts on specific task. We sincerely appreciate the reviewer’s valuable suggestion—this is indeed an important direction, and we are already planning finer-grained expert-level analyses in our ongoing work.
>
> #### Regarding the explanation for "MUI indicates better generalization lacks justification"
>
> - **Response 2.2:** We kindly refer the reviewer to the General Response for a detailed explanation. At the same time, we emphasize that measuring the generalization capability of MoE models remains a non-trivial open problem.  The absence of a rigorous and universally valid metric for generalization is precisely what makes current evaluation challenges difficult—and it is also the motivation for our extensive efforts to explore alternative, mechanism-based perspectives such as MoE-MUI. For this reason, we would greatly appreciate any additional detailed suggestions from you that could help us further strengthen or refine this conclusion.
>
> #### Regarding the concern "just measuring the routing sparsity"
>
> - **Response 2.3:** Our observations are **not** attributable to routing sparsity. In our preliminary experiments, we found that with as few as 200 samples from GSM8K, the selected MoE model (eg. GPT-OSS-20b) already activates over 99% of its experts, the same extent as the neuron, and other models exhibit similar behavior. So the sparsity of  MUI reflects the model’s internal activation patterns, rather than an artifact of routing sparsity.
>
> #### Regarding the suggestions for Figure 1 and model naming
>
> - **Response 2.4:** Thank you very much for your helpful suggestions. We have updated Figure 1 and the model naming in the revised version accordingly and sincerely welcome any further feedback.

---

> ### Author Response · Authors · 2025-11-24
> **Response to Reviewer SNTy**
>
> Dear Reviewer SNTy,
>
> Thank you once again for your review and comments. We kindly request your feedback and comments, especially regarding whether our response addressed your concerns. Your feedback is crucial to us. We apologize for any inconvenience caused.

---

> ### Author Response · Authors · 2025-11-28
> **Response to Reviewer SNTy**
>
> Dear Reviewer,
>
> Thank you again for your review.
> **As the rebuttal/discussion period is drawing to a close**, we respectfully request your attention to our response. We believe the points raised in our rebuttal effectively address your initial concerns.
> If you find that our responses have clarified your concerns, we would be grateful for any final comments or a re-evaluation of your score.
> We appreciate your valuable time and input.

---

### Official Review · Reviewer_Lumt · 2025-11-03

**Soundness:** 3
**Presentation:** 3
**Contribution:** 2
**Rating:** 4
**Confidence:** 3

**Summary:**

This paper adapts the Model Utilization Index (MUI) to analyze internal behaviors of MoE models, revealing how neuron and expert activation patterns evolve with training and correlate with generalization. The authors perform large-scale experiments on 13 MoE models and OLMoE checkpoints, finding that higher-performing models use fewer neurons and exhibit more expert collaboration, especially in architectures with shared experts.

**Strengths:**

The paper goes beyond the conventional benchmark-centric evaluation and tackles the open question of how MoE models utilize their enormous capacity. This focus on interpretability and internal mechanisms is timely and original.

The study is thorough, examining 13 different MoE models spanning several popular families (GPT-OSS, Qwen3, DeepSeek, etc.)

**Weaknesses:**

The proposed MUI, while useful, is not a fundamentally new class of interpretability method; it is a well-engineered adaptation. There is no novel learning algorithm, architectural change, or optimization scheme proposed—only a diagnostic and analysis framework. The conclusions are similar to Cao et al. (2025) that observed a similar relationship between lower MUI and stronger generalization

The authors describe reduced neuron usage as “a reflection of stronger generalization”. However, the paper does not directly measure generalization in a rigorous way (e.g., out-of-distribution performance, or an experiment like “take a model, improve its generalization via some regularization, see MUI drop” or similar).Thus, while the interpretation is plausible and supported by prior work’s Utility Law, it isn’t conclusively demonstrated.

**Questions:**

One question is how MoE-MUI compares to MUI in dense models (from Cao et al. 2025) or even whether MoE models utilize neurons more efficiently than dense counterparts of similar performance. For instance, when saying GPT-OSS (MoE) has strikingly low MUI, it would be insightful to know: low compared to what? Perhaps a dense GPT-style model with similar capability uses a larger fraction of its neurons for the same tasks.

How exactly did you measure the degree of expert cooperation?

---

> ### Author Response · Authors · 2025-11-19
> **Response to Reviewer Lumt (part 1)**
>
> Thank you for your review and suggestions. It appears there may have been some misunderstandings regarding the work and contribution. Here is our response to your questions.
>
> ### **General Response**
>
> Thank you for recognizing that this work explores a novel and promising direction. Measuring the generalization or capability of MoE models is an inherently challenging task—especially as model ability improves with scale. Beyond the risk of in-domain contamination in public benchmarks, it is fundamentally difficult to use finite test data to faithfully reflect a model’s unbounded generalization space. Nevertheless, given the importance of model evaluation, we believe it is worthwhile to explore alternative perspectives beyond performance metrics. Our work takes a first step by analyzing MoE models through their internal mechanisms, leveraging multiple interpretability techniques to understand how neuron/expert utilization relate to evaluation outcomes. As with any new approach, we acknowledge that controversy and limitations are inevitable. **We kindly ask the reviewers to consider whether, based on our detailed analyses, our reasoning is internally consistent and whether this line of inquiry opens up meaningful and promising research directions**—even if it is not possible to exhaustively address all aspects within a single paper.
>
> Our central claim—lower MUI and better performance indicates better generalization—is supported from multiple perspectives:
>
> 1. **Model analysis within the same series:** When analyzing models with clearly different intrinsic capacities within the same family, we observe that stronger MoE models consistently achieve higher benchmark performance while exhibiting lower MUI values. In particular, DeepSeek-Coder-V2-Lite is further pre-trained from a checkpoint of DeepSeek-V2-Lite, which provides a clean comparison: the performance gap between the two, combined with the widespread deployment and adoption of the former, as suggested by its higher download counts, offers a practical and training-driven demonstration of this trend.
> 2. **Supporting evidence from prior work:** Similar patterns have been observed in controlled and dense-model settings. Anthropic (2023) [1] showed that as a single ReLU network improves its generalization, its parameters become increasingly specialized. Cao et al. (2025) [2] reported consistent findings for dense models.
>
> Taken together, these findings suggest that MUI serves as an indicator of intrinsic model capacity and generalization.
>
> [1] Superposition, Memorization, and Double Descent
>
> [2] Model Utility Law: Evaluating LLMs beyond Performance through Mechanism Interpretable Metric

---

> ### Author Response · Authors · 2025-11-19
> **Response to Reviewer Lumt (part 2)**
>
> #### Regarding the **misunderstanding** of our contribution "only a diagnostic and analysis framework" and "not a fundamentally new class of interpretability method":
>
> - **Response 1.1:** Our goal is **not** to propose a new class of interpretability methods. On the contrary, we intentionally use multiple existing interpretability techniques to analyze MoE models and address the evaluation challenges that cannot be resolved by performance-only metrics.  In this work we adopt multiple interpretability methods (see Section 2.2 for the main setting and Section 4 and Appendix C for additional implementation) and extend the MUI to account for MoE-specific characteristics, such as routing mechanisms and the distinction between shared and routed experts, as also highlighted by Reviewer SNTy, which is non-trivial and technically meaningful.  Please refer the **General Response** for the more details of ours contributions.
>
> #### Regarding the explanation for "reduced neuron utilization with model evolution"
>
> - **Response 1.2:**  Considering the limitation of performance-only evaluation—one of them is the difficulty of rigorously defining what constitutes out-of-domain—our work examines how MUI and performance evolve with increasing MoE capacity/generalization from several perspectives detailed in the **General Response.**
>
>   On the other hand, measuring MoE models' capability/generalization itself remains a non-trivial open question—if there existed a rigorous and universally valid way to quantify generalization, many of today’s evaluation challenges would already be resolved. Our work aims to shed light on this issue by investigating the intrinsic mechanisms of MoE models. For this reason, we would greatly appreciate any additional detailed suggestions from you that could help us further strengthen or refine this conclusion.
>
>   Finally, we appreciate the reviewer’s suggestion regarding training-based analyses. Beyond the model-comparison evidence presented in general response, we have already begun related training experiments to explore this direction more deeply. But given the scope of this submission, we do not elaborate on them here.
>
> #### Regarding the question "how MoE-MUI compares to MUI in dense models" and "GPT-OSS has strikingly low MUI,"
>
> - **Response 1.3:** As a metric reflecting intrinsic MoE behaviors, directly comparing MUI values between MoE and dense architectures is not appropriate, given their fundamentally different structures and activation patterns. The observation that GPT-OSS exhibits a relatively low MUI refers specifically to comparisons within MoE architectures.
>
> #### Regarding the clarification of "expert cooperation"
>
> - **Response 1.4:** Here refers to the proportion of key experts activated for the given task, as defined in Equation (6). If this phrasing caused any ambiguity, we will clarify it explicitly in the revised version of the paper.

---

> ### Author Response · Authors · 2025-11-24
> **Response to Reviewer Lumt**
>
> Dear Reviewer Lumt,
>
> Thank you once again for your review and comments. We kindly request your feedback and comments, especially regarding whether our response addressed your concerns. Your feedback is crucial to us. We apologize for any inconvenience caused.

---

> ### Author Response · Authors · 2025-11-28
> **Response to Reviewer Lumt**
>
> Dear Reviewer,
>
> Thank you again for your review.
> **As the rebuttal/discussion period is drawing to a close**, we respectfully request your attention to our response. We believe the points raised in our rebuttal effectively address your initial concerns.
> Specifically, regarding the scope of our contribution and the methodology, we clarified that our work focuses on MoE analysis using established techniques rather than defining a new interpretation method.
> If you find that our responses have clarified your concerns, we would be grateful for any final comments or a re-evaluation of your score.
> We appreciate your valuable time and input.

---

### Author Response · Authors · 2025-12-01
**Rebuttal Summary & Final Remarks**

Dear Reviewers and ACs,

Thanks for your time and effort in reviewing our paper. Here, we briefly summarize the rebuttal status:

We are glad that the reviewers recognize the contribution of our work and its future potential, specifically highlighted by Reviewer SNTy—that measuring the generalization or capability of MoE models is an inherently challenging task, especially as model ability improves with scale and goes beyond the risk of in-domain contamination in public benchmarks. Our work takes a first step by analyzing MoE models through their internal mechanisms, leveraging multiple interpretability techniques to understand how neuron/expert utilization relates to evaluation outcomes. Overall, our reasoning is internally consistent and supported by results across different model familes, providing finer-grained insights.

We believe we have addressed all the reviewers' concerns:

- Regarding Reviewers [Lumt, W8fg], who expected us to propose new principled interpretability methods or further analyze existing techniques: this was not the target of our work. Furthermore, considering that interpretability is a rapidly evolving field with currently no consensus on how to definitively determine encoded neuron features, we employed multiple existing techniques to facilitate our MoE analysis.

- Regarding Reviewers [Lumt, SNTy, W8fg, fTfs], who suggested exploring more through training: we are pleased that everyone recognizes the guiding significance and huge potential of our work for training. In the paper, we have already provided clear analysis regarding model iteration and different training versions (e.g., DeepSeek-V2-Coder $\to$ DeepSeek-V2 & OLMoE); however, full pre-training experiments are beyond the scope of our resources.

- Regarding Reviewers [Lumt, SNTy, W8fg] suggest to measure generalization in a rigorous way: the fact is that the very lack of rigorous definitions is what causes the unreliability challenges in model evaluation, and this paper represents the first step toward addressing that.
- Regarding Reviewer [W8fg], who requested "concrete, quantitative decision rules" for MoE-MUI, we have provided a detailed explanation in Response 3.3.
- We have also optimized the figures in the revised manuscript based on suggestions from Reviewers [fTfs, SNTy].

Regrettably, there was no further engagement from the reviewers during the discussion period, which is particularly unfortunate given that Reviewers [Lumt, W8fg] appear to have misunderstandings regarding the paper, and some of the inquiries were not clearly articulated. We hope the AC will consider our detailed responses.

Best regards,

The Authors

---

### Note · Authors · 2026-01-31

I have read and agree with the venue's withdrawal policy on behalf of myself and my co-authors.

---

### Meta-Review · Area_Chair_vxSc · 2026-01-06

**Summary:**

In this paper, the authors study Mixture-of-Experts (MoE) models from an internal-mechanism perspective by extending the Model Utilization Index (MUI) to account for routing and expert-level behaviors. The core contribution is an empirical analysis across multiple MoE model families that correlates neuron utilization, training dynamics, and expert collaboration with benchmark performance.

The reviews present a mixed assessment of the paper. While reviewers appreciate the novelty of moving beyond benchmark-centric evaluation and the scale of the empirical study, most reviewers agree that the paper suffers from the following key weaknesses:

(1) *Insufficient evidence linking MUI to generalization*: Multiple reviewers raise concerns that the paper overstates the link between lower MUI and stronger generalization. In particular, Reviewer SNTy notes that the argument assumes newer or higher-performing models necessarily generalize better, which leads to circular reasoning. Similarly, Reviewer W8fg points out the lack of controlled experiments or independent measures of generalization, so the evidence remains largely correlational and influenced by architectural and training differences.

(2) *Interpretability but not sufficiently mechanistic*: Several reviewers point out that, despite framing the work as an investigation of internal mechanisms, the analysis remains largely descriptive and statistical. Reviewer SNTy notes that while expert collaboration is reported, the paper does not analyze what individual experts or neurons actually learn, nor whether they encode complementary or semantically meaningful features. Reviewer W8fg further notes that identifying "key neurons" does not clearly show what these neurons actually do, so the interpretability claims remain high-level rather than explanatory. Consequently, the paper falls short of providing a mechanistic explanation for why lower MUI should correspond to improved capability or efficiency, rather than merely observing activation patterns.

The authors have attempted to address the reviewers' concerns during the rebuttal period by clarifying the scope of the contribution and positioning MUI as a diagnostic metric rather than a causal or predictive one. However, these clarifications do not fully resolve the reviewers' concerns, and the core issues above remain present in the paper.

Therefore, I am inclined toward rejection. I encourage the authors to strengthen future submissions by moderating the generalization claims, incorporating controlled or theoretically grounded analyses, and providing more fine-grained mechanistic insights into expert and neuron specialization.

**Reviewer Concerns:**

Please refer to the summary.

**Reviewer Scores:**

Please refer to the summary.

---

### Decision · Program_Chairs · 2026-01-26

Reject